# An efficient PDE based method to compute pressure boundary conditions in regional geodynamic models

Anthony Jourdon[1,2] and Dave A. May[1]

[1]Institute of Geophysics and Planetary Physics, Scripps Institution of Oceanography, UC San Diego, La Jolla, CA, USA
[2]Department of Earth and Environmental Sciences, Ludwig-Maximillians-Universität München, Munich, Germany

**Correspondence:** Anthony Jourdon (jourdon.anthon@gmail.com)

**Abstract.** Modelling the pressure in the Earth's interior is a common problem in Earth sciences. In this study we propose a method based on the conservation of momentum of a fluid reduced to the case of a hydrostatic scenario or a uniformly moving fluid to approximate the pressure. This results in a partial differential equation (PDE) which can be solved using the classical numerical methods. In hydrostatic cases, the computed pressure is the lithostatic pressure. In non-hydrostatic cases, we show that this PDE based approach better approximates the total pressure than the classical 1D depth-integrated approach. To illustrate the performance of this PDE based formulation we present several hydrostatic and non-hydrostatic 2D models in which we compute the lithostatic pressure or an approximation of the total pressure respectively. Moreover, we also present a 3D rift model which uses that approximated pressure as a time-dependent boundary condition to simulate far-field normal stresses. This model shows a high degree of non-cylindrical deformation resulting from the stress boundary condition that is accommodated by strike-slip shear zones. We compare the result of this numerical model with a traditional rift model employing free-slip boundary conditions to demonstrate the first order implications of considering "open" boundary conditions in 3D thermo-mechanical rift models.

## 1 Introduction

In Earth sciences and geodynamic modelling, computing the pressure can be essential. Specifically, numerous regional thermo-mechanical studies use the lithostatic pressure or a reference pressure based on some density structure as a normal stress boundary condition (e.g., Baes et al., 2018; Brune, 2014; Brune et al., 2012, 2014, 2017; Chertova et al., 2012, 2014; Glerum et al., 2018; Ismail-Zadeh et al., 2013; Popov and Sobolev, 2008; Quinteros et al., 2010; Yamato et al., 2008). By imposing only the normal stress, material is permitted to flow in and out of the domain in response to the other boundary conditions and or deformation in the domain interior. This is inherently closer to the reality of the dynamics within a regional segment of the Earth, compared to a regional domain which is closed and in which neither inflow/outflow is permitted. Hence, the ultimate intent of imposing the normal stress is to provide dynamical behaviour which is similar to that which would occur if the models were performed in a much larger domain. Moreover, the reference pressure can also be used as an initial guess for the pressure when solving linear or non-linear systems flow problems with iterative methods.

The common approach to compute a reference pressure $P$ is to define a set of depth columns and integrate the rock density $\rho(\boldsymbol{x})$ over each column to obtain the pressure at depth. That is, to compute the pressure $P$ at some point $\boldsymbol{x}'$, we evaluate the 1D integral

$$P(\boldsymbol{x}') = P_s + \int_{\boldsymbol{x}'_s}^{\boldsymbol{x}'} \rho(\boldsymbol{x}) \|\boldsymbol{g}(\boldsymbol{x})\| \, dx, \tag{1}$$

where $\boldsymbol{x}'_s$ is the projection of $\boldsymbol{x}'$ onto the surface of the Earth in the direction opposite to the gravity vector $\boldsymbol{g}$ and $P_s$ is the reference pressure at the surface $\boldsymbol{x}'_s$. For the case of a constant density and gravity, this expression reduces to $P(\boldsymbol{x}') = P_s + \rho g D$ where $D$ is the distance (depth) given by $D = \|\boldsymbol{x}'_s - \boldsymbol{x}'\|$ and $g = \|\boldsymbol{g}\|$. When the density is a function of space and gravity is constant, the 1D integral is decomposed into different segments $D_i$ and a suitable quadrature rule is applied over each segment. For example using a 1-point Gauss quadrature rule we have

$$P(\boldsymbol{x}') = P_s + \sum_i \int_{D_i} \rho(\boldsymbol{x}) g \, dx \approx P_s + \sum_i \rho_i g \, D_i, \tag{2}$$

where $\rho_i$ is the density at the centroid of the segment $D_i$. For the case of a uniform mesh with cell edges aligned with the gravity vector, all the cell edges / vertices are located along straight lines which are parallel to the direction of gravity. Hence Eq. (2) can be simply evaluated by traversing along a column associated with a set of cells (or vertices). In this special case, the sub-division of the integral is naturally defined by mesh cells. If the column sweep is performed from the surface to depth, then only a single pass over each cell in a column is required to compute the pressure at any depth within that column by accumulating values from cells at shallower depths. That is if we traverse from segments $i = 0, 1, 2, \ldots, N$ where the segments are ordered such that $D_{i+1}$ is located at greater depth than $D_i$, then we have the following sequence $P_0 = P_s + \rho_0 g D_0$, $P_1 = P_s + \rho_0 g D_0 + \rho_1 g D_1 = P_0 + \rho_1 g D_1, \ldots, P_N = P_s + \sum_i^N \rho_i g D_i = P_{N-1} + \rho_N g D_N$.

Although evaluating Eq. (2) may appear simple, its implementing may be inefficient, or algorithmically complex for general use. Below we outline some common use-cases which render the column-wise integration difficult (or expensive):

1. A mesh with cell edges (2D) or faces (3D) which are not aligned with the gravity vector (Figure 1a);

2. An unstructured mesh (Figure 1b);

3. A density structure (or gravity vector) which is spatially varying;

4. A parallel decomposition of the mesh (Figure 1c);

5. Time-dependence in the density, or mesh coordinates which requires continual re-evaluation of the reference pressure.

To compute $P(\boldsymbol{x}')$ we first have to define the location $\boldsymbol{x}'_s$. In general this is non-trivial for use-cases 1. and 2. If both the density and gravity are constant, then the only complexity associated with meshes identified in points 1. and 2. relate to computing $\boldsymbol{x}'_s$. Due to the fact that the path of the integral (i.e. the "column") does not coincide in general with a set of mesh cells or vertices, the line integral must be performed for each point $\boldsymbol{x}'$ in the mesh - the single pass approach used in the gravity

aligned mesh is not possible. If the density (or gravity) vary in space throughout the domain, the integral must be approximated via a suitable sub-division in space and, or a quadrature rule. Assuming that the density is a piece-wise constant over each cell, the simplest approximation would be to determine the intersection between the line segment $[\boldsymbol{x}', \boldsymbol{x}'_s]$ and each cell, and apply a 1-point quadrature rule over the intersecting segment times. In Figure 1a, b we depict the complexity of this procedure for a non-coordinate aligned and an unstructured mesh. When performing simulations in parallel where the mesh is distributed across multiple MPI ranks, even for the case of a uniform mesh aligned with the gravity vector the column-wise integration approach is somewhat complicated. Individual MPI ranks may compute their local contribution to the sum of accumulated pressures, however the final pressure requires a partial sum to be performed over mesh sub-domains which intersect the 1D line integral. The global reduction (taken MPI ranks overlapping with each 1D line integral) is complicated to define for mesh types identified in points 1. and 2. Lastly, if the reference pressure associated with some density structure is to be used as a boundary condition in a mechanical model, time dependence of that density structure (or mesh) will require one to re-compute the reference pressure at each time step. Hence the efficiency of the implementation used to compute the pressure is important.

Moreover, when the density structure evolves with time as deformation occurs, the pressure gradients may no more be aligned with the gravitational acceleration vector. In these non-hydrostatic cases, this pressure is not lithostatic. However, to be able to provide an approximation for the total pressure or to use stress boundary conditions, it is still important to approximate the total pressure in the best possible way.

For these reasons, we propose an efficient, mesh and numerical method (finite elements, finite differences, finite volumes, *etc*...) independent way to compute a reference pressure associated with the density structure of a domain in hydrostatic cases or an approximation of the total pressure for non-hydrostatic cases for all scenarios 1-5 above by solving a partial differential equation (PDE) derived from the conservation of the non-inertial momentum equation for an incompressible fluid. We also present thermo-mechanical numerical models and static numerical models applied to Earth sciences and geodynamics to show the usefulness of this approach.

## 2   PDE based pressure formulation

For an incompressible fluid in a domain $\Omega$, the non-inertial form of the conservation of momentum is given by the Stokes equation

$$\nabla \cdot \boldsymbol{\tau} - \nabla P + \rho \boldsymbol{g} = \boldsymbol{0}, \tag{3}$$

where $\boldsymbol{v}$ is the velocity of the fluid, $P$ is the total pressure, $\boldsymbol{\tau} = 2\eta \dot{\boldsymbol{\varepsilon}}(\boldsymbol{v})$ the deviatoric stress tensor with $\eta$ the viscosity and $\dot{\boldsymbol{\varepsilon}}(\boldsymbol{v})$ the strain rate tensor, $\rho := \rho(\boldsymbol{x}, t)$ is the density and $\boldsymbol{g} := \boldsymbol{g}(\boldsymbol{x}, t)$ is the gravity vector and $\boldsymbol{x}, t$ denote the space and time respectively. The incompressibility constraint is given as

$$\nabla \cdot \boldsymbol{v} = 0. \tag{4}$$

In the context of our problems we will decompose the boundary of the domain into two non-overlapping segments: $\partial\Omega_{\text{surf}}$ that we will regard as the free surface and prescribe that tangential and normal stresses are zero, i.e. $\boldsymbol{\tau} - P\mathbb{I} = \boldsymbol{0}$ and $\partial\Omega_{\text{i}}$ which

denotes the interior parts of the boundary along which we may impose any valid combination of velocity / stress in the normal and tangential directions. Furthermore, $\partial\Omega = \partial\Omega_{\mathrm{i}} \cup \partial\Omega_{\mathrm{surf}}$ and $\partial\Omega_{\mathrm{i}} \cap \partial\Omega_{\mathrm{surf}} = \emptyset$. The outward pointing unit normal vector to $\partial\Omega$ will be denoted via $\hat{\boldsymbol{n}}$.

To define the pressure associated with the density structure we make the *ansatz* that $\boldsymbol{v} = \boldsymbol{0}$, hence Eq. (4) is trivially satisfied and Eq. (3) reduces to the usual hydrostatic equilibrium problem

$$\boldsymbol{0} = -\nabla P + \rho\boldsymbol{g}. \tag{5}$$

For spatial dimensions $n_d = 2, 3$, Eq. (5) is over-determined as there are more equations ($n_d$) than unknowns:

$$\begin{bmatrix} \frac{\partial}{\partial x} \\ \frac{\partial}{\partial y} \\ \frac{\partial}{\partial z} \end{bmatrix} P = \rho(\boldsymbol{x}) \begin{bmatrix} g_x \\ g_y \\ g_z \end{bmatrix}. \tag{6}$$

As such, there is no unique solution to Eq. (5). To obtain a unique solution to Eq. (5) we require a single equation for $P$. This can be achieved by taking the divergence of Eq. (5):

$$\nabla \cdot \nabla P = \nabla \cdot (\rho\boldsymbol{g}). \tag{7}$$

Eq. (7) will be referred to as the Pressure Poisson equation (PPE).

Eq. (7) can be obtained in an alternative manner with less restrictive assumptions. First, we assume that we have a solution $(\boldsymbol{v}, p)$ for Eqs. (3),(4). Then we take the divergence of the momentum equation (as before) and integrate over $\Omega$

$$\int_\Omega \nabla \cdot (\nabla \cdot \boldsymbol{\tau})\, dV - \int_\Omega \nabla \cdot \nabla P\, dV + \int_\Omega \nabla \cdot (\rho\boldsymbol{g})\, dV = 0. \tag{8}$$

Then, applying the divergence theorem to the first term we obtain

$$\int_{\partial\Omega} (\nabla \cdot \boldsymbol{\tau}) \cdot \hat{\boldsymbol{n}}\, dS - \int_\Omega \nabla \cdot \nabla P\, dV + \int_\Omega \nabla \cdot (\rho\boldsymbol{g})\, dV = 0. \tag{9}$$

If we assume the boundary term on the LHS is small, we have

$$\int_\Omega \nabla \cdot \nabla P\, dV \approx \int_\Omega \nabla \cdot (\rho\boldsymbol{g})\, dV \tag{10}$$

which must be true for any arbitrary domain, thus resulting in Eq. (7).

Assuming that $\hat{\boldsymbol{n}}$ is constant or slowly varying and using that $\boldsymbol{\tau}$ is symmetric yields

$$(\nabla \cdot \boldsymbol{\tau}) \cdot \hat{\boldsymbol{n}} = n_i \frac{\partial}{\partial x_j} \tau_{ij} \approx \frac{\partial}{\partial x_j}(\tau_{ji} n_i) = \nabla \cdot (\boldsymbol{\tau}\hat{\boldsymbol{n}}). \tag{11}$$

Hence dropping the first term in (9) is equivalent to saying that $\boldsymbol{\tau}\hat{\boldsymbol{n}} \approx \boldsymbol{0}$ for all $\boldsymbol{x} \in \partial\Omega$. Alternatively, the term is zero if $\nabla \cdot \boldsymbol{\tau} = \boldsymbol{0}$ for all $\boldsymbol{x} \in \partial\Omega$. This condition is satisfied if the fluid experiences either rigid body translations, or rigid body rotation along $\partial\Omega$.

## 2.1 Boundary conditions

A unique solution to Eq. (7) requires boundary conditions to be specified on $P$. Our choice of boundary conditions for Eq. (7) are motivated by Earth-like bodies. The boundary conditions will be specified in the usual manner, i.e. in terms of a Dirichlet constraint in which we impose $P$ and Neumann constraints in which we impose the behaviour of $\nabla P \cdot \hat{\boldsymbol{n}}$.

Along the surface of the domain which represents the free-surface of the Earth, we impose

$$P = 0 \quad \text{for all } \boldsymbol{x} \in \partial\Omega_{\text{surf}}. \tag{12}$$

Eq. (12) is a Dirichlet constraint and defines that the reference (or datum) pressure should be zero on the surface of our geological body. This is consistent with the observation that the mean pressure on all points on the surface (above sea level) are approximately equal. This Dirichlet boundary condition is a natural extension of the free surface boundary condition used for the flow problem in Eq. (3),(4), namely $\hat{\boldsymbol{n}} \cdot (\boldsymbol{\tau} - P\mathbb{I})\hat{\boldsymbol{n}} = \hat{\boldsymbol{n}} \cdot (\boldsymbol{\tau} - P\mathbb{I})\hat{\boldsymbol{t}} = 0$, which reduces to $\hat{\boldsymbol{n}} \cdot \boldsymbol{\tau}\hat{\boldsymbol{n}} - P = 0$ and $\hat{\boldsymbol{n}} \cdot \boldsymbol{\tau}\hat{\boldsymbol{t}} = 0$ with $\hat{\boldsymbol{t}}$ a tangent unit vector to the boundary such that $\hat{\boldsymbol{n}} \cdot \hat{\boldsymbol{t}} = 0$. Eq. (12) is consistent with a fluid at rest since $\boldsymbol{\tau} = \boldsymbol{0}$. In the non-hydrostatic case, we require that $\boldsymbol{\tau}\hat{\boldsymbol{n}} \approx 0$ to arrive at Eq. (12). We also note that $\hat{\boldsymbol{n}} \cdot \boldsymbol{\tau}\hat{\boldsymbol{n}}$ is proportional to the mean curvature $\kappa$ of the boundary $\partial\Omega$ (Barth and Carey, 2007). Hence if there is zero topography $\kappa = 0$, and $P = 0$ on $\partial\Omega_{\text{surf}}$, or if the change in topography is small, then $\kappa \approx 0$ and $P \approx 0$.

Two different boundary conditions are introduced to constrain $\nabla P \cdot \hat{\boldsymbol{n}}$. These are defined as a direct extension of the 1D hydrostatic assumptions to 2D and 3D domains. We first introduce some additional quantities which will aid the definition of the Neumann boundary condition. First, we split $\partial\Omega_i$ into two parts, such that $\partial\Omega_i = \partial\Omega_\perp \cup \partial\Omega_\parallel$. Next we define the gravity unit vector $\hat{\boldsymbol{g}}$ such that:

$$\boldsymbol{g} = g\hat{\boldsymbol{g}} \tag{13}$$

and the unit vector $\hat{\boldsymbol{g}}_\perp$ which is perpendicular to $\hat{\boldsymbol{g}}$, i.e.

$$\hat{\boldsymbol{g}} \cdot \hat{\boldsymbol{g}}_\perp = 0. \tag{14}$$

The first constrain states that $P$ should increase only in the direction of gravity. Hence, from Eq. (5) we have

$$\nabla P \cdot \hat{\boldsymbol{g}} = \rho g \|\hat{\boldsymbol{g}}\|^2 = \rho g, \qquad \text{for all } \boldsymbol{x} \in \partial\Omega_\parallel. \tag{15}$$

The second constraint states that $P$ does not change along directions perpendicular to the gravity

$$\nabla P \cdot \hat{\boldsymbol{g}}_\perp = 0, \qquad \text{for all } \boldsymbol{x} \in \partial\Omega_\perp. \tag{16}$$

Since the normal to the boundary $\partial\Omega$ unit vector $\hat{\boldsymbol{n}}$ can be decomposed according to

$$\hat{\boldsymbol{n}} = (\hat{\boldsymbol{n}} \cdot \hat{\boldsymbol{g}})\hat{\boldsymbol{g}} + (\hat{\boldsymbol{n}} \cdot \hat{\boldsymbol{g}}_\perp)\hat{\boldsymbol{g}}_\perp, \tag{17}$$

we have that

$$\nabla P \cdot \hat{\boldsymbol{n}} = (\hat{\boldsymbol{n}} \cdot \hat{\boldsymbol{g}})\nabla P \cdot \hat{\boldsymbol{g}} + (\hat{\boldsymbol{n}} \cdot \hat{\boldsymbol{g}}_\perp)\nabla P \cdot \hat{\boldsymbol{g}}_\perp. \tag{18}$$

Hence the two Neumann boundary conditions may be stated as

$$\nabla P \cdot \hat{\boldsymbol{n}} = (\hat{\boldsymbol{n}} \cdot \hat{\boldsymbol{g}}) \rho g + (\hat{\boldsymbol{n}} \cdot \hat{\boldsymbol{g}}_\perp) \nabla P \cdot \hat{\boldsymbol{g}}_\perp \qquad \text{for all } \boldsymbol{x} \in \partial\Omega_\parallel \tag{19}$$

and

$$\nabla P \cdot \hat{\boldsymbol{n}} = (\hat{\boldsymbol{n}} \cdot \hat{\boldsymbol{g}}) \nabla P \cdot \hat{\boldsymbol{g}} \qquad \text{for all } \boldsymbol{x} \in \partial\Omega_\perp. \tag{20}$$

Constraints (19) and (20) may appear peculiar since both the LHS and RHS involve the gradient of pressure. In principal, to obtain a unique solution to Eq. (7) one can constrain the gradient in any direction, independent of the boundary normal $\hat{\boldsymbol{n}}$ – our 1D inspired gradients do exactly that.

We note for domains with boundaries which are parallel to either $\hat{\boldsymbol{g}}$ or $\hat{\boldsymbol{g}}_\perp$ then the Neumann conditions (19) and (20) simplify and in some cases do not provide any information to constrain $\nabla P \cdot \hat{\boldsymbol{n}}$. For example, consider a 2D Cartesian domain right / left boundaries $\Gamma_{r,l}$ with normal $\hat{\boldsymbol{n}} = (\pm 1, 0)$, and a bottom boundary $\Gamma_b$ with normal $\hat{\boldsymbol{n}} = (0, -1)$ and $\hat{\boldsymbol{g}} = g(0, -1)$. Invoking Eq. (19) we obtain

$$\nabla P \cdot \hat{n} = \begin{cases} \nabla P \cdot \hat{\boldsymbol{g}}_\perp = \nabla P \cdot \hat{\boldsymbol{n}} & \text{for } \boldsymbol{x} \in \Gamma_{l,r} & \text{(21a)} \\ \rho g & \text{for } \boldsymbol{x} \in \Gamma_b . & \text{(21b)} \end{cases}$$

Invoking Eq. (20) we obtain

$$\nabla P \cdot \hat{n} = \begin{cases} 0 & \text{for } \boldsymbol{x} \in \Gamma_{l,r} & \text{(22a)} \\ \nabla P \cdot \hat{\boldsymbol{g}} = \nabla P \cdot \hat{\boldsymbol{n}} & \text{for } \boldsymbol{x} \in \Gamma_b . & \text{(22b)} \end{cases}$$

Clearly conditions (21a) and (22b) are redundant. As such, the usage of constraints (19) and (20) cannot be used arbitrarily.

One may also consider employing both (19) and (20) simultaneously. In this way we would obtain

$$\begin{aligned} \nabla P \cdot \hat{\boldsymbol{n}} &= (\hat{\boldsymbol{n}} \cdot \hat{\boldsymbol{g}}) \nabla P \cdot \hat{\boldsymbol{g}} + (\hat{\boldsymbol{n}} \cdot \hat{\boldsymbol{g}}_\perp) \nabla P \cdot \hat{\boldsymbol{g}}_\perp \\ &= (\hat{\boldsymbol{n}} \cdot \hat{\boldsymbol{g}}) \rho g + (\hat{\boldsymbol{n}} \cdot \hat{\boldsymbol{g}}_\perp) 0 \\ &= \rho g \hat{\boldsymbol{g}} \cdot \hat{\boldsymbol{n}} = \rho \boldsymbol{g} \cdot \hat{\boldsymbol{n}}, \end{aligned} \tag{23}$$

which is identical to the result obtained by computing the dot product of Eq. (5) with $\hat{\boldsymbol{n}}$. Constraint (23) certainly avoids the potential issue of using constraints (19) and (20). If we again consider the 2D Cartesian domain example, imposing (23) on all of $\partial\Omega_i$ is equivalent to imposing (21b) and (22a).

Nevertheless, for an arbitrarily shaped domain, using boundary conditions (15), (16) or (23) does not yield the same result (see section 3.3). For a general use (i.e. when considering arbitrarily shaped domains) we suggest to employ Eq. (15) & (16) as they are a direct extension of the 1D hydrostatic assumptions to 2D and 3D domains.

## 2.2 Weak formulation

To define the weak formulation of the PPE we will use functions that are square-integrable in the sense of Lebesgue, i.e.

$$L_2(\Omega) := \left\{ u : \Omega \to \mathbb{R} \;\middle|\; \int_\Omega u^2 \, dV < \infty \right\}$$

and functions from the $H_1(\Omega)$ Sobolev space

$$H_1(\Omega) := \left\{ u : \Omega \to \mathbb{R} \quad \middle| \quad u, \nabla u \in L_2(\Omega) \right\}.$$

Finally we will require the space of functions in $H_1(\Omega)$ which vanish on the Dirichlet boundary $\partial\Omega_{\text{surf}}$:

$$H_1^d(\Omega) = \left\{ u \in H_1(\Omega) \quad \middle| \quad u = 0 \text{ on } \partial\Omega_{\text{surf}} \right\}.$$

Given a test function $q \in H_1^d(\Omega)$ the weak form of the PPE is obtained by multiplying Eq. (7) by $q$ and integrating both sides over $\Omega$

$$\int_\Omega q \nabla \cdot \nabla P \, dV = \int_\Omega q \nabla \cdot (\rho \boldsymbol{g}) \, dV. \tag{24}$$

Applying integration by parts to the left and right hand sides yields

$$\int_\Omega \nabla q \cdot \nabla P \, dV - \int_{\partial\Omega_i} q \nabla P \cdot \hat{\boldsymbol{n}} \, dS = \int_\Omega \nabla q \cdot (\rho \boldsymbol{g}) \, dV - \int_{\partial\Omega_i} q \rho \boldsymbol{g} \cdot \hat{\boldsymbol{n}} \, dS. \tag{25}$$

Note that the boundary $\partial\Omega_{\text{surf}}$ does not appear in Eq. (25) since the test function $q$ vanishes along the Dirichlet boundary. We also note that Eq. (25) only requires $\rho \boldsymbol{g} \in L_2(\Omega)$, thus the formulation is valid for cases when the density $\rho$ is discontinuous.

Splitting the surface integrals over the two segments $\partial\Omega_i = \partial\Omega_\perp \cup \partial\Omega_\parallel$ and using Eqs. (19), (20) we have

$$\int_\Omega \nabla q \cdot \nabla P \, dV - \int_{\partial\Omega_\parallel} q \left[ (\hat{\boldsymbol{n}} \cdot \hat{\boldsymbol{g}}) \rho g + (\hat{\boldsymbol{n}} \cdot \hat{\boldsymbol{g}}_\perp) \nabla P \cdot \hat{\boldsymbol{g}}_\perp \right] dS - \int_{\partial\Omega_\perp} q \left[ (\hat{\boldsymbol{n}} \cdot \hat{\boldsymbol{g}}) \nabla P \cdot \hat{\boldsymbol{g}} \right] dS = \int_\Omega \rho g \nabla q \cdot \hat{\boldsymbol{g}} \, dV - \int_{\partial\Omega_i} q \rho g \hat{\boldsymbol{g}} \cdot \hat{\boldsymbol{n}} \, dS. \tag{26}$$

Noting that the second term of the LHS and part of the last term on the RHS exactly cancel yields the following

$$\int_\Omega \nabla q \cdot \nabla P \, dV - \int_{\partial\Omega_\parallel} q \left[ (\hat{\boldsymbol{n}} \cdot \hat{\boldsymbol{g}}_\perp) \nabla P \cdot \hat{\boldsymbol{g}}_\perp \right] dS - \int_{\partial\Omega_\perp} q \left[ (\hat{\boldsymbol{n}} \cdot \hat{\boldsymbol{g}}) \nabla P \cdot \hat{\boldsymbol{g}} \right] dS = \int_\Omega \rho g \nabla q \cdot \hat{\boldsymbol{g}} \, dV - \int_{\partial\Omega_\perp} q \rho g \hat{\boldsymbol{g}} \cdot \hat{\boldsymbol{n}} \, dS. \tag{27}$$

From Eq. (25), the weak formulation obtained if using Eq. (23) applied over all of $\partial\Omega_i$ is simply

$$\int_\Omega \nabla q \cdot \nabla P \, dV = \int_\Omega \rho g \nabla q \cdot \hat{\boldsymbol{g}} \, dV. \tag{28}$$

## 2.3 Implementation

The strong (Eq. (7)) and weak (Eq. (25)) formulations of the pressure Poisson problem can be solved using the standard spatial discretisation techniques, e.g. finite differences or finite elements. Moreover, since the equation is of Poisson type, it is readily amenable to be solved using standard iterative multigrid and/or direct solvers. Lastly, because the formulation is expressed in terms of a PDE, it is also straightforward to compute the pressure on parallel computing architecture as we can re-use existing discretisation implementations that support domain decomposition.

## 3 Numerical examples

In this section we provide several numerical models to show:

1. the accuracy and consistency of the method for hydrostatic cases,

2. the accuracy of the approximation of the total pressure in non-hydrostatic cases,

3. the effect of using the depth integrated approach and the pressure Poisson problem approach to impose boundary conditions to the momentum equation,

4. the usefulness of the method for 3D geodynamics thermo-mechanical modelling.

### 190  3.1  Hydrostatic pressure

To compare the numerical solution of Eq. (7) with analytical solution obtained with Eq. (2) we designed four hydrostatic models (in 1D and 2D) for which the analytical solution is easily obtained (Figure 2 and Figure 3).

#### 3.1.1  Box domain

We define the domain $\Omega = x \in [0,1] \times y \in [0,1]$ and assume that $\boldsymbol{g} = (0,-1)$. We consider three depth dependent density
structures $\rho = \rho(y)$ which thus admit a hydrostatic pressure solution, i.e. satisfies $\partial P/\partial x = 0$, $\partial P/\partial y = -\rho(y)g$.

**Case 1:** Assumes a constant density, $\rho(y) = 1$ (Figure 2a,b). The analytic pressure solution is given by

$$P(y) = \int_{y}^{y_s=1} \rho g \, dy = [\rho g y]_y^{y_s} = \rho g(y_s - y) = y_s - y = 1 - y. \tag{29}$$

**Case 2:** Assumes a continuous depth varying density $\rho(y) = 2 - y$ (Figure 2c,d). The analytic pressure solution is given by

$$P(y) = \int_{y}^{y_s=1} \rho g \, dy = \int_{y}^{y_s=1} g(2-y) \, dy = \left[ g\left(-\frac{1}{2}y^2 + 2y\right)\right]_y^{y_s} = g\left(\frac{1}{2}\left(y^2 - y_s\right) + 2(y_s - y)\right) = \frac{1}{2}y^2 - 2y + \frac{3}{2}. \tag{30}$$

**Case 3:** A discontinuous density such that $\rho(y) = 1$ for $y \in [0.5,1]$ and $\rho(y) = 2$ for $y \in [0,0.5)$ (Figure 2e,f). The analytic pressure solution is given by

$$P(y) = \begin{cases} \rho(y)g(1-y) = (1-y) & \forall y \geqslant 0.5 \\ \rho(y \geqslant 0.5)gD + \rho(y)g(0.5-y) = D+1-2y & \forall y < 0.5 \end{cases} \tag{31}$$

with $D$ the distance between the surface and the $y$ coordinate at which the pressure is computed.

A finite element (FE) method employing an unstructured triangular mesh with a $P_2$ function space was used to obtain the
numerical solution for each case. The FE method was applied to Eq. (27) using boundary conditions described by Eq. (15) at

the base and Eq. (16) on the lateral sides. Along the upper surface we impose the Dirichlet constraint $P = 0$. Unless otherwise stated, when solving the PPE with FEs, the Dirichlet constraints are imposed strongly (i.e. point-wise) whilst Neumann constraints are imposed weakly via surface integrals defined on facets of the FE cells which live on the boundary of the domain. Accordingly, all points living on $\partial\Omega_{\text{surf}}$ (including corner points) will be associated with Dirichlet constraint. Figure 2a-f shows the 1D and 2D solution of these three models. On the 1D models both the numerical and analytical solutions of Eq. (7) and Eq. (2) respectively are shown while on the 2D models only the numerical solution is provided. Since the $P_2$ FE approximation contains the monomials 1, $y$ and $y^2$, the FE solution exactly reproduces the analytic solution for case 1 and case 2 – independent of the number of finite elements used in the domain (e.g. sub-dividing the box into two triangles would be sufficient to obtain an exact solution). For case 3, the analytic pressure solution is piecewise linear, hence provided the density discontinuity is exactly resolved by the faces of the triangular FE mesh (which was the case here), the FE method exactly reproduces the analytic solution.

### 3.1.2 Half annulus domain

The 2D half annulus model aims to show the efficiency of the method to compute the lithostatic pressure in a body with a radial gravity vector and concentric density structure (Figure 3a). This model represents a domain extending from

$$
\begin{cases}
\theta \in [-\frac{\pi}{2}, \frac{\pi}{2}], \\
r \in [R_e - 2891, R_e] \text{ km}
\end{cases}
\tag{32}
$$

where $\theta$ is the polar angle and r the radius in polar coordinates, $R_e$ is the approximative Earth radius (6371 km) and $R_e - 2891$ km is the approximative core-mantle boundary. Mapped into Cartesian coordinates gives $x = r\sin(\theta)$ and $y = r\cos(\theta)$. The gravity vector pointing to the centre ($\boldsymbol{x} = (0,0)$) is defined as $\boldsymbol{g} = -9.8 \left( \frac{x}{\sqrt{x^2+y^2}}, \frac{y}{\sqrt{x^2+y^2}} \right)$ m.s$^{-2}$ and the density is defined as 5 concentric layers with a constant density in each (Figure 3a). The pressure was computed by solving Eq. (27) with the boundary condition of Eq. (15) on the core-mantle boundary and the conditions of Eq. (16) on the sides parallel to $\hat{\boldsymbol{g}}$. At the surface of the domain we impose the Dirichlet constraint $P = 0$. As in Sec. 3.1 we use a finite element discretisation employing an unstructured mesh of triangles employing a $P_2$ function space. Pressure values vary from 0 to 180 GPa with a concentric distribution following the density distribution and the gravity vector orientation (Figure 3b).

The numerical solution extracted at $x = 0$ along a line parallel to the gravity vector field reproduces the analytical solution computed for a 1D profile using Eq. (2) for the density distribution displayed in the half annulus model (Figure 3c). The difference between the pressure obtained using the depth integrated approach and the PPE is very small (Figure 3d). Unlike the analytic solution for the box models, the analytic solution for $P$ here is non-polynomial (in Cartesian coordinates) hence the FE solution (which was discretised in Cartesian coordinates) cannot exactly reproduce the analytic solution.

The four hydrostatic models clearly illustrate that the solutions obtained using the depth integrated approach and the pressure Poisson equation (with one set of boundary constraints) are equivalent for scenarios which admit a hydrostatic solution.

## 3.2 Non-hydrostatic pressure

Here, we show the differences and accuracy of the depth integrated (Eq. (2)) and the pressure Poisson equation (Eq. (7)) approaches to approximate the total pressure. First, we compute and compare the pressure from the different methods in a large domain (referred to as the "global" model) containing a topographic perturbation (Figure 4). Then, we compute the pressure in a smaller domain (referred to as the "regional" model) and show the accuracy of the different methods to approximate the total pressure from the large domain (Figures 5a,c and 6). Finally, we show the velocity field resulting from applying these approximated pressure as boundary conditions to solve the conservation of momentum in the small domain (Figure 5b,d). The pressure Poisson problem was discretised and solved using the same FE method described in Sec. 3.1. The flow field was computed using the same underlying FE mesh, using a mixed $P_2$-$P_1$ function space for velocity and pressure respectively.

### 3.2.1 "Global" model

We define a large domain $\Omega_G = x \in [-10, 10] \times y \in [0, 1]$ representing a "global" model i.e. 20 times larger than the domain of interest in order to avoid boundary conditions influence (Figure 4). In this domain, we introduce a topographic perturbation through a slope defined as $y_s(x) = -\frac{1}{4}x$ where $y_s$ is the surface between $x = 0$ and $x = 1$. For demonstration purpose this topography is highly exaggerated with respect to the depth of the domain compared with the actual regional geodynamic models. Moreover, we use a constant density $\rho = 1$ and a vertical gravity vector $\boldsymbol{g} = (0, -1)$.

We solve the flow problem described by Eqs. (3), (4) using a constant viscosity and density, along with the following boundary conditions: no-slip at base, free-slip on right and left faces and a free-surface along the top of the model domain. Due to the topography, a non-vertical pressure gradient that will drive flow is generated below that perturbation (Figure 4a). The generated flow shows a velocity field characteristic of a gravitational collapse. Figure 4b shows the difference between the total pressure solution from Eq. (3) with the pressure $P_d$ obtained by solving Eq. (7) using the boundary conditions described by Eq. (16) on the vertical sides, Eq. (15) on the bottom boundary and $P_d = 0$ on the upper surface. Figure 4c shows the difference with the pressure $P_a$ obtained with the Eq. (2).

The difference between the total pressure and the approximated pressure $P_d$ is negligible in the non-perturbed domain and increases with the topography perturbation. It shows that as the system tends towards a hydrostatic state the total pressure and the approximated pressure $P_d$ tends to the same value i.e. the hydrostatic pressure. Yet, in the vicinity of the topographic pertur-bation the differences between the total pressure and the approximated pressure can be extremely small (Figure 4b). In contrast, the difference between the total pressure and depth integrated approximated pressure $P_a$ is larger below the topographic slope, particularly below the points at which the slope begins and ends (Figure 4c). In general, the pressure $P_a$ obtained by applying the 1D solution is less accurate than $P_d$ within both the interior and along the left / right / bottom boundaries.

### 3.2.2 "Regional" model

Nevertheless, modelling a domain 20 times larger than the domain of interest can hardly be achieved in practice mainly due to the numerical cost it represents. Thus, the boundary conditions are a first order component of regional models in order to

best capture the global behaviour and interactions in a region without having to model the whole Earth. Therefore, we define a smaller domain $\Omega_R = x \in [0,1] \times y \in [0,1]$ representing a "regional" model (Figure 5). This domain represents the portion of the large domain in which the topographic slope is defined.

In this case we aim to apply a normal stress on the boundaries of our regional model that will generate a flow similar (or close) to the flow generated in the "global" model. To achieve this we present two models using Eq. (7) and Eq. (2) respectively to compute an approximated pressure that is then used as a boundary condition for the momentum equation (3) as:

$$\boldsymbol{\sigma n} = -P_\alpha \hat{\boldsymbol{n}}, \quad \alpha = d, a \tag{33}$$

on vertical boundaries, where $P_\alpha = \{P_d, P_a\}$ denotes the pressure computed using the pressure Poisson equation or 1D depth integrated approach respectively. The viscous flow problem for the regional domain setting employs the following additional boundary conditions: free-surface on the top boundary and a no-slip condition at the base. To solve for the pressure $P_d$ using Eq. (7) we impose the following boundary conditions: $P_d = 0$ on the surface, $\nabla P_d \cdot \hat{\boldsymbol{n}} = (\hat{\boldsymbol{n}} \cdot \hat{\boldsymbol{g}}) \nabla P_d \cdot \hat{\boldsymbol{g}}$ on vertical sides and $\nabla P_d \cdot \hat{\boldsymbol{n}} = (\hat{\boldsymbol{n}} \cdot \hat{\boldsymbol{g}}) \rho g + (\hat{\boldsymbol{n}} \cdot \hat{\boldsymbol{g}}_\perp) \nabla P_d \cdot \hat{\boldsymbol{g}}_\perp$ at the bottom.

Figure 5a shows the pressure field $P_d$ computed on the regional domain and the total pressure $P$ extracted from the global model. The approximated pressure $P_d$ highlights differences with the total pressure from the global model especially in depth along the boundaries (Figure 6b, d - blue lines). These differences are mainly due to the size of the domain that defines only the perturbed region without providing informations about the domain in which it is enclosed. The boundary condition described by Eq. (16) used to solve Eq. (7) enforces that the pressure gradient must be co-linear with the gravity vector on the boundary. Therefore, given the definition of $\boldsymbol{g}$, $\nabla P$ is enforced to be vertical on the vertical sides while the deflection of the pressure field due to the topographic perturbation should occur in spatial offset from the topographic perturbation as shown by the total pressure in the global model (Figure 4a).

In Figure 5b we show the velocity field resulting from solving the Stokes equation (Eqs. (3), (4)) using $P_d$ in Eq. (33). The flow field highlights velocities of around $5 \times 10^{-2}$ (velocity unit) at the surface pointing toward the right side of the domain, i.e. the bottom of the slope. Velocities progressively decrease at depth. While the orientation of the velocity field is more laminar than in the global model, its magnitude is very close, with the highest velocities being only $1.5 \times$ higher than in the global model.

Figure 5c displays the pressure field $P_a$ computed from the depth integrated approach (Eq. (2)) within the regional domain and the total pressure $P$ extracted from the global model. Because of the 1D behaviour of that method, the differences between the approximated pressure $P_a$ and the total pressure $P$ (Figures 5c and 6b, d - red lines) are independent of the domain size and boundary conditions, thus they are exactly the same than in the global model (Figure 4c). The velocity field (Figure 5d) resulting from using these approximated pressure values as a boundary condition to solve the momentum equation are approximatively 4 times higher than velocities obtained while using the pressure $P_d$ (Figure 5b) and approximatively 6 times higher than in the global model (Figure 4a). As for the velocity field orientation, the vectors at the top of the slope show that the material uplifts while in the global model velocities at the same location show a gravitational sliding. This orientation results from a too high stress value imposed in the boundary conditions by the value of the pressure $P_a$ on boundaries (Figure 6).

These simple tests demonstrate that the approximated pressure $P_d$ computed from the Eq. (7) is more accurate to approximate the total pressure $P$ than the pressure $P_a$ computed from the Eq. (2). The only area where this is not true, is located in the bottom left corner of the regional model, again due to the boundary condition not capturing the deflection of the pressure due to the size of the domain. Thus, to be used as boundary conditions for the momentum equation, the pressure computed with the pressure Poisson problem should be preferred. Moreover, as the domain size increases the error with the total pressure decreases which is not the case with the depth integrated approach due to its 1D nature.

### 3.3 Influence of the boundary condition type

The boundary conditions used to solve the pressure Poisson problem are stated in Eqs. (15), (16), (23). To show the influence of the boundary conditions choices on the resulting pressure field, we define an irregular quadrilateral domain with the same topographic perturbation than the one described in Section 3.2 with a constant density $\rho = 1$, a gravity vector $\boldsymbol{g} = (0, -1)$ and a Dirichlet boundary condition $P_d = 0$ at the top surface ($\Gamma_s$). The irregular domain (shown in Figure. 7) is constructed such that none of the three boundary segments defining $\partial\Omega_i = \Gamma_1 \cup \Gamma_2 \cup \Gamma_3$ are parallel or perpendicular to $\boldsymbol{g}$. The PPE was solved using the same FE method described in Sec. 3.1.

In Figure 7 we show different pressure fields $P_d$ obtained using several different boundary condition configurations. These results, show that the boundary condition described by Eq. (16) (or Eq. (20)) force the pressure gradient to be parallel to $\boldsymbol{g}$, i.e. $\partial_x P_d = 0$ (Figure 7 panel b segment $\Gamma_3$; panel c segments $\Gamma_{1,3}$; panel d segments $\Gamma_{1,2}$). We also confirm that boundary condition described by Eq. (15) (or Eq. (19)) constrain $\nabla P_d$ to be equivalent to the 1D solution of Eq. (2) on the boundary (Figure 7 panel a segments $\Gamma_{1,2,3}$; panel b segments $\Gamma_{1,2}$; panel c segment $\Gamma_2$; panel d segment $\Gamma_3$). Figure 7a highlights that the solution of the PPE can be identical to that obtained using the depth integrated approach with a specific choice of boundary conditions.

Moreover, Figure 8a shows the pressure field $P_d$ obtained using the boundary condition described by Eq. (23). The pressure solution is relatively similar to the solution obtained in Figure 7c. However, as shown on Figure 8b the solution along the boundary (and therefore in the interior also) differs since Eq. (23) does not enforce the pressure gradient to be vertical on the boundary, compared with the condition (16).

As previously noted, discretisations employing the weak form with constraint (23) are certainly simpler to implement than imposing the constraints (19), (20) as all the surface integrals cancel (see Eq. (28)), and thus no surface integrals appear in either the linear form or bilinear form. We also re-iterate that when the domain has boundaries which are aligned with $\boldsymbol{g}$, using the boundary conditions (15) on the boundaries orthogonal to $\boldsymbol{g}$ and the condition (16) on the boundaries which are parallel to $\boldsymbol{g}$ result in an identical formulation.

### 3.4 Thermo-mechanical model

#### 3.4.1 Physical model

To simulate the long term evolution of the deformation of the lithosphere we solve the stationary, non-inertial form of the conservation of momentum described by Eq. (3) with the incompressible constraint (Eq. (4)). Moreover, to consider the temperature variations in the domain the time dependent conservation of energy is solved

$$
\rho_0 C_p \left( \frac{\partial T}{\partial t} + \boldsymbol{v} \cdot \nabla T \right) = \nabla \cdot (k \nabla T) + H \tag{34}
$$

with $T$ the temperature, $t$ the time, $k$ the thermal conductivity, $H$ the heat source, $\rho_0$ the reference density and $C_p$ the thermal heat capacity.

The numerical solution of Eqs. (3) & (4) is obtained using a mixed finite element method which independently discretises the velocity and pressure fields. Hence the numerical velocity and pressure obtained are solutions of the weak form of the Stokes problem, given by

$$
\mathcal{A}(\boldsymbol{w}, \boldsymbol{v}) + \mathcal{B}(\boldsymbol{w}, p) + \mathcal{B}(\boldsymbol{v}, q) - \int_{\Gamma_N} \boldsymbol{w} \cdot \boldsymbol{T}(v, p) \, dS = - \int_{\Omega} \rho \boldsymbol{w} \cdot \boldsymbol{g} \, dV
$$

where $\boldsymbol{w} \in H_1(\Omega)$ and $q \in L_2(\Omega)$ are test functions for the velocity and pressure respectively, $\Gamma_N$ denotes the Neumann boundary, $\boldsymbol{T}$ denotes the traction vector given by $\boldsymbol{T}(v, p) = (\boldsymbol{\tau}(\boldsymbol{u}) - p\mathbb{I})\hat{\boldsymbol{n}}$ with $\hat{\boldsymbol{n}}$ being the outward pointing normal vector from the boundary. The bilinear forms for the Stokes problem are given by (Elman et al., 2014):

$$
\mathcal{A}(\boldsymbol{w}, \boldsymbol{v}) = \int_{\Omega} 2\eta \dot{\boldsymbol{\varepsilon}}(\boldsymbol{w}) : \dot{\boldsymbol{\varepsilon}}(\boldsymbol{v}) \, dV, \quad \dot{\boldsymbol{\varepsilon}}(\boldsymbol{v}) = \tfrac{1}{2}[\nabla \boldsymbol{v} + (\nabla \boldsymbol{v})^T],
$$

$$
\mathcal{B}(\boldsymbol{v}, q) = - \int_{\Omega} q \nabla \cdot \boldsymbol{v} \, dV.
$$

Both the Stokes and thermal problem were solved using the parallel finite element code `pTatin3D` (May et al., 2014, 2015), which employs a mixed $Q_2$-$P_1$ discretisation for velocity and pressure.

#### 3.4.2 Initial conditions and rheology

To model the strain localization we use nonlinear visco-plastic rheologies expressed in term of viscosity. The ductile parts of the domain are simulated using an Arrhenius flow law for dislocation creep

$$
\eta_v = A^{-\frac{1}{n}} \left( \dot{\varepsilon}^{II} \right)^{\frac{1}{n}-1} \exp \left( \frac{Q + PV}{nRT} \right), \tag{35}
$$

where $A$, $Q$ and $n$ are material-defined parameters (see Table 1), $R$ is the universal gas constant, $V$ the activation volume and $\dot{\varepsilon}^{II}$ the square root of the strain rate second invariant computed as

$$
\dot{\varepsilon}^{II} = \sqrt{\frac{1}{2} \dot{\varepsilon}_{ij} \dot{\varepsilon}_{ij}}. \tag{36}
$$

The brittle parts of the domain are simulated using a Drucker-Prager yield criterion adapted to continuum mechanics, given by

$$\eta_p = \frac{C\cos(\phi) + P\sin(\phi)}{2\dot{\varepsilon}^{II}},$$ (37)

with $C$ the cohesion of the material and $\phi$ the friction angle.

The modelled domain contains 4 initial flat layers representing the upper continental crust, the lower continental crust, the lithosphere mantle and the asthenosphere mantle respectively (Figure 9a). The upper crust extends from the surface of the domain ($y = 0$ km) to $y = -25$ km and is modelled with a dislocation creep quartz rheology (Ranalli, 1997). The lower crust extends from $y < -25$ km to $y = -35$ km and is modelled with a dislocation creep anorthite rheology (Rybacki and Dresen, 2000). The lithosphere mantle extends from $y < -35$ km to $y = -120$ km while the asthenosphere mantle extends from $y < -120$ km to $y = -450$ km. They are both modelled using a dislocation creep olivine flow law (Hirth and Kohlstedt, 2003).

The initial density distribution follows the lithologies and is reported in Table 1. In addition, the density varies with pressure and temperature following the Boussinesq approximation

$$\rho(P,T) = \rho_0(1 - \alpha(T - T_0) + \beta(P - P_0)),$$ (38)

with $\rho_0$ the reference density at $T_0$ and $P_0$, $P$ is the total pressure computed from the conservation of momentum (Eq. (3)) and continuity equation (Eq. (4)), $\alpha$ the thermal expansion and $\beta$ the compressibility. The Boussinesq approximation states that perturbations of density, if sufficiently small, can only be considered in the buoyancy term and neglected elsewhere regardless of the origin of the perturbation.

Moreover, the initial temperature field is computed as a steady-state solution of the heat equation

$$\nabla \cdot (k\nabla T) + H = 0,$$ (39)

using a surface temperature of $T = 0°$C at $y = 0$ km and $T = 1450°$C at $y = -450$ km. Moreover, to simulate an adiabatic thermal gradient in the asthenosphere due to thermal convection, the initial temperature field is solved with a conductivity of $k = 70$ W.m$^{-1}$.K$^{-1}$ in the asthenospheric mantle. However, for the actual model run we used a more realistic conductivity of $k = 3.3$ W.m$^{-1}$.K$^{-1}$ to solve Eq. (34). Other thermal parameters are reported in Table 1

### 3.4.3 Boundary conditions

To show the influence of the normal stress boundary condition we compare two rift models. In the reference model, an extension velocity of $v_x = 1$cm.yr$^{-1}$ is applied on the whole faces of normal $x$, whilst on faces of normal $z$ a free-slip boundary condition is applied (Figure 9c). To ensure mass conservation we impose an inflow velocity on the bottom face of normal $y$ to balance any outflow which occurs due to the imposed extension. Along the surface of the model we use a free surface (zero normal stress, zero tangential stress) boundary condition.

**Table 1.** Physical parameters for the thermo-mechanical rift model.

| Parameter | Units | Upper crust | Lower crust | Lithosphere mantle | Asthenosphere mantle |
|---|---|---|---|---|---|
| $A$ | $\text{MPa}^{-n}.\text{s}^{-1}$ | $6.7 \times 10^{-6}$ | $13.4637$ | $2.5 \times 10^{4}$ | $2.5 \times 10^{4}$ |
| $n$ | - | $2.4$ | $3.0$ | $3.5$ | $3.5$ |
| $Q$ | $\text{kJ.mol}^{-1}$ | $156$ | $345$ | $532$ | $532$ |
| $\phi$ | $^\circ$ | $30$ | $30$ | $30$ | $30$ |
| $C$ | $\text{MPa}$ | $20$ | $20$ | $20$ | $20$ |
| $V$ | $\text{m}^{3}.\text{mol}^{-1}$ | $0$ | $38 \times 10^{-6}$ | $8 \times 10^{-6}$ | $8 \times 10^{-6}$ |
| $C_p$ | $\text{m}^{2}.\text{K}^{-1}.\text{s}^{-2}$ | $850$ | $850$ | $850$ | $850$ |
| $k$ | $\text{W.m}^{-1}.\text{K}^{-1}$ | $2.7$ | $2.85$ | $3.3$ | $3.3$ |
| $H$ | $\mu\text{W.m}^{-3}$ | $1.5$ | $0.3$ | $0$ | $0$ |
| $\rho_0$ | $\text{kg.m}^{-3}$ | $2700$ | $2850$ | $3300$ | $3300$ |
| $\alpha$ | $\text{K}^{-1}$ | $3 \times 10^{-5}$ | $3 \times 10^{-5}$ | $3 \times 10^{-5}$ | $3 \times 10^{-5}$ |
| $\beta$ | $\text{Pa}^{-1}$ | $10^{-11}$ | $10^{-11}$ | $10^{-11}$ | $10^{-11}$ |

The second rift model (Figure 9b) uses the same Dirichlet boundary conditions on faces of normal $x$. On the faces of normal $z$ we impose a Neumann boundary condition as

$$\boldsymbol{T} = -P_d\,\hat{\boldsymbol{n}}, \tag{40}$$

where $P_d$ is the pressure computed with Eq. (7) and $\hat{\boldsymbol{n}}$ the normal vector pointing outward the domain.

To account for the density evolution through time due to the deformation and material advection, Eq. (7) is solved at every non-linear iteration for each time step and the Neumann boundary condition described by Eq. (40) is evaluated at every non-linear iteration. Using $\rho(P,T)$ computed from Eq. (38) to evaluate the pressure $P_d$ going into the boundary condition described by Eq. (40) adds a new non-linearity to the system.

The bottom of the domain is prescribed an inflow condition balancing the outflow and the surface of the domain is a free surface where the mesh deforms accordingly to the computed velocity field. These Neumann boundary conditions allow material to flow both in and out through the boundary depending only on the Dirichlet boundary conditions and deformation which occurs inside the modelled domain.

### 3.4.4 Pressure Poisson problem in the 3D geodynamic model

In the context of our finite-element forward model, we also solve the pressure Poisson problem using finite elements. As such, to compute $P_d$ we employ the weak formulation given by Eq. (27), using boundary conditions from Eq. (15) and Eq. (16) on bottom boundary and vertical boundaries respectively. In our particular implementation, we employ $Q_1$ for $P_d$, and these $Q_1$ elements overlap the $Q_2$ elements used to approximate the velocity.

As a demonstration of the computed $P_d$ using this approach, in Figures 10c and 10d we show the approximated pressure in our rift model at 8.7 Myr after large deformations that led to mantle exhumation, differential thinning of the continental crust and a variable topography has occurred. In this model, $P_d$ was evaluated on a mesh consisting of $256 \times 64 \times 128$ $Q_1$ finite elements on 1024 MPI ranks. The discrete pressure Poisson system was solved using geometric multigrid. As a rough estimate, solving for $P_d$ required $\sim 0.2\%$ of the time required to solve the non-linear viscous flow problem. Obviously this value is strongly dependent on both the physical model (linear viscous versus non-linear viscous) and the implementation details, efficiency of how the discrete flow problem is solved. However, when considering even the simplest flow problem imaginable (i.e. linear iso-viscous flow laws), it does remain true that solving the Poisson problem will be far less expensive than solving either the linear or non-linear viscous flow problem.

### 3.4.5 Tectonics evolution

The model using free-slip boundary conditions displays a cylindrical deformation pattern that could be reduced to a two dimensional model. As shown by the shear zones orientation and strain regime, the deformation is only extensional and perpendicular to the extension direction (Figure 11a to 11d and Figure 12). This strain localization is directly due to the free-slip boundary condition stating that any flow perpendicular to the boundary is prohibited.

On the contrary, the model using the $P_d$ pressure as a boundary conditions displays a non-cylindrical deformation. While extensional shear zones perpendicular to the extension direction develop in the central part of the domain, the edges of the rift experience oblique and strike-slip deformation (Figure 11e to 11h). As the extension goes on, the extensional deformation localizes along a spreading centre causing an increasing inflow on the boundaries of the domain with the normal stress boundary condition (Figure 13c). As a result, near these boundaries the velocity field introduces non-cylindrical features which are accommodated by strike-slip faults (Figure 11g, h). These strike-slip faults delimit a triangular region terminating on a triple junction between two strike-slip faults and a ridge (RFF triple junction). Along these strike-slip faults, the deformation is partitioned between purely vertical strike-slip shear zones and shallow dipping normal shear zones rooting into the strike-slip shear zones (Figure 12).

## 4 Discussion

### 4.1 Alternative PDE based approaches

Recall that the starting point of defining the PPE was purely algebraic, with the sole intention of removing the non-uniqueness associated with Eq. (5). Here we discuss two alternative PDE based approaches constructed with a similar rational.

Rather than enforcing constraint (15) only along the boundary, suppose we wished to enforce it everywhere throughout the domain,

$$\nabla P \cdot \hat{\boldsymbol{g}} = \rho g, \quad \text{for } \boldsymbol{x} \in \Omega. \tag{41}$$

This constraint can be interpreted as the steady-state solution of the following scalar hyperbolic PDE:

$$\frac{\partial P}{\partial \tau} + \hat{\boldsymbol{g}} \cdot \nabla P = \rho g \tag{42}$$

where $\tau$ plays the role of a time-like parameter having units of length and $\hat{\boldsymbol{g}}$ plays the role of a velocity-like quantity. Along the "inflow segments" $\Gamma_{in} = \{\boldsymbol{x} \in \partial\Omega : \hat{\boldsymbol{g}} \cdot \hat{\boldsymbol{n}} < 0\}$ we will impose $P = 0$.[1] On the outflow segments $\Gamma_{out} = \partial\Omega \setminus \Gamma_{in}$, due to the hyperbolic nature of the PDE, no boundary constraint are required. Since we seek the steady-state solution of (42), no initial

condition is required but for completeness we chose $P(\tau = 0) = 0$. Hence, the solution of Eq. (42) is equivalent to solving

$$\frac{dP}{d\tau} = \rho g \tag{43}$$

along the family of characteristics given by

$$\frac{d\boldsymbol{x}}{d\tau} = \hat{\boldsymbol{g}}, \tag{44}$$

with $P(\tau = 0) = 0$ and $\boldsymbol{x}(\tau = 0) \in \Gamma_{in}$.

Compared to the PPE, the hyperbolic formulation has several disadvantages.

1. We have less freedom to specify how $\nabla P$ varies along the boundary. The choice of BCs is largely dictated by the "inflow" / "outflow" segments. Along outflow segments, the only constraint available is Eq. (41) evaluated on $\partial\Omega$. If "inflow" occurs on any part of $\partial\Omega$ not contained in $\partial\Omega_{\text{surf}}$, we have to choose a flux BC as using $P = 0$ does not make physical sense. Consistency may require one uses a constraint which is independent of the PDE, e.g. Eq. (16).

2. The formulation may place restrictions on the shape of $\partial\Omega$. If we wish to avoid the definition of new flux BCs (described above), the domain must be defined such that for every $\boldsymbol{x}_b \in \partial\Omega \setminus \partial\Omega_{\text{surf}}$, there exists a characteristic which intersects both $\boldsymbol{x}_b$ and $\partial\Omega_{\text{surf}}$.

3. The lack of flexibility in controlling the boundary behaviour of $\nabla P$ will in general result in solutions of (42) being identical to a family of 1D solutions to Eq. (2) applied in directions parallel to the direction of gravity. See Figure 14b,

445    d.

4. The spatial discretisation required for the accurate solution of (42) are arguably more complicated to implement (on unstructured meshes) compared with discretisations for the Poisson equation. Scalable multi-level solvers for the steady-state hyperbolic problem are much more challenging to develop cf. the Poisson problem.

From a linear algebra perspective, the non-uniqueness of Eq. (5) can alternatively be addressed by (i) discretising Eq. (5) in

space yielding $\mathbf{G}\mathbf{p} = \mathbf{F}$, and then (ii) solving the normal equations

$$\mathbf{G}^T\mathbf{G}\tilde{\mathbf{p}} = \mathbf{G}^T\mathbf{F}. \tag{45}$$

---

[1]"Inflow" in the context of (42) can be thought of as the origin of "information" which enters the physical domain, whilst the $\hat{\boldsymbol{g}}$ defines the direction in which this information travels.

This approach obtains a unique solution which minimizes the following objective function

$$\|\mathbf{G}\tilde{\mathbf{p}} - \mathbf{F}\|_2.$$

In some sense, this approach is like the discrete counterpart of the PPE, in so much as $\mathbf{G}^T\mathbf{G}$ being the discrete Laplacian defined on the approximation space used to represent pressure. The similarity between the pressure solution from the PPE and $\tilde{\mathbf{p}}$ are shown in Figure 14a, c. Similar to the hyperbolic formulation, obtaining the pressure via the normal equations is restrictive from a modelling perspective as the formulation does not allow to control the behaviour of $\nabla P$ on the boundary, only the Dirichlet data on $\partial\Omega_{\mathrm{surf}}$ can be specified. In Figure 14a, c we provide snapshots of the pressure obtained from solving (45) in two different domains. Interestingly, the solutions indicate that $\nabla P \cdot \hat{\boldsymbol{g}}$ appears to be approximately zero along the boundary, despite the lack of a constraint enforcing this.

### 4.2 Flexibility of the PPE with respect to boundary conditions

Since the PPE is a second order PDE, the formulation permits a range of possible boundary constraints on $\nabla P$ to be imposed. Specific choices allow one to define whether (i) the equivalent of 1D pressure profiles as would be obtained by applying Eq. (2) along a boundary face, or (ii) an approximation to the pressure field on the boundary which would be obtained if a complete flow field was computed in a much larger "global" domain.

Experiments showed that the PPE approach better approximates the total pressure computed from the momentum equation. Therefore, its use as a boundary condition (or as an initial guess) for the pressure field to solve the momentum equation is preferred over hydostatic type solutions associated with Eq. (2). Moreover, as the domain size increases, the PPE formulation gives a more accurate approximation of the total pressure than the 1D depth integrated approach.

### 4.3 Implication on lithosphere deformation

In the geodynamic rift model using the pressure computed with Eq. (7) as a boundary condition produces a velocity field perpendicular to the extension direction in the rift axis (Figure 13). This velocity field introduces non-cylindrical deformation accommodated by oblique and strike-slip structures (Figure 11). The results of this study are very similar to previous studies directly applying an inflow perpendicularly to the extension direction (Le Pourhiet et al., 2018; Jourdon et al., 2020). At the tip of the rift, a triangular region delimited by strike-slip faults or very oblique rift develops to accommodate the oblique velocity field. The similarity of these results shows that using open boundary conditions instead of kinematic boundary conditions in 3D may reveal first order implications for the lithosphere strain localization. In the case of geodynamic systems presenting the characteristics of a propagating rift (or ridge) with oblique and strike-slip deformation at its tip, considering the forces applied by the surrounding material weight could be the first order process at the origin of non-cylindrical deformation.

### 4.4 Linear and non-linear density

To compute the pressure using Eq. (7) two approaches can be considered. The first approach (also the simplest) is to consider that the density $\rho$ is defined as a reference density $\rho_0$ that only depends on rock type for a reference state *e.g.* $T_0$, $P_0$. In that

case, Eq. (7) is linear and only depends on the reference density structure. The second approach is to consider that the density $\rho$ can vary with respect to other parameters. In our geodynamic example we considered that the density can vary with pressure and temperature according to Eq. (38). In that case the Eq. (7) becomes non-linear. This kind of non-linearity is not rare in geodynamics and the formulation of Eq. (38) may appear in the pressure computation when solving for an incompressible

Boussinesq approximation or a compressible Stokes problem (e.g., Dannberg and Heister, 2016; King et al., 2010; Tackley, 2008).

## 5   Conclusions

In this study we presented a method to compute a reference pressure associated with the density structure of a domain in which we cast the problem in terms of a partial differential equation (PDE). From a practical standpoint, the PDE approach is

generic (it is applicable to all spatial discretisation and on any type of computational grid), efficient and applicable in parallel computing environments. From the modelling perspective, the PDE approach has specific advantages for example in models with a variable density structure (stationary or time-dependent), and models which employ a reference pressure as a boundary condition of the flow problem (stationary or time-dependent). Re-evaluating that pressure in time-dependent problems is not problematic (even if the mesh deforms) since solving the Poisson problem can be performed using optimal preconditioners

(e.g. geometric or algebraic multigrid). Importantly, the time to solve the pressure Poisson problem is a small fraction of the time required to solve the linear (or non-linear) incompressible viscous flow problem. Moreover, we also demonstrate that in non-hydrostatic cases, the PDE formulation results in a better approximation of the total pressure than the 1D depth integrated approach.

Lastly, we showed in the context of 3D geodynamic models of continental rifting, that using a reference pressure as a

boundary condition within the flow problem resulted in non-cylindrical velocity fields. These 3D velocity fields produced strain localization in the lithosphere along large scale strike-slip shear zones and the formation and evolution of triple junctions.

*Code availability.* The code `pTatin3D` used in this study to produce the 3D thermo-mechanical models is an open source free software licensed under GPL3. The supplementary repository contains the version of the code used to produce the models presented in this study. To run the same models, user should use the driver named `test_driver_checkpoint_fv.app` and the options files (`.opts`) provided

in the supplementary files. We also provide `Firedrake` code (Balay et al., 2019, 1997; Dalcin et al., 2011; Rathgeber et al., 2016) to compute the pressure Poisson problem in a half annulus domain, in a deformed domain and in the large and small domains with a topography perturbation used in this study. The version of `Firedrake` used is `0.13.0+4944.g22178416` and is freely available. We also provide `FEniCS` code (Alnaes et al., 2015; Logg et al., 2012b; Logg and Wells, 2010; Logg et al., 2012a) to reproduce the models solving for the normal and hyperbolic equations. The version of `FEniCS` used is `2016.1.0` and is freely available.

*Author contributions.* Project conceptualisation: AJ, DAM. Mathematical development: AJ, DAM. Software development: AJ, DAM. Conceived and designed the experiments: AJ, DAM. Performed the experiments: AJ, DAM. Analysed the results: AJ. Drafted the manuscript: AJ, DAM. Procured funding, project supervision: DAM.

*Competing interests.* There are no competing interests

*Acknowledgements.* This project was supported by NSF Award EAR-2121666. The authors gratefully acknowledge the Gauss Centre for
Supercomputing e.V. (www.gauss-centre.eu) by providing computing time on the GCS Supercomputer SuperMUC-NG at Leibniz Supercomputing Centre (www.lrz.de) through project pr63qo. AJ acknowledges funding from the European Union's Horizon 2020 research and innovation programme (TEAR ERC Starting grant no. 852992). The authors also thanks C. Thieulot and R. Gassmöeller for their constructive reviews.

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

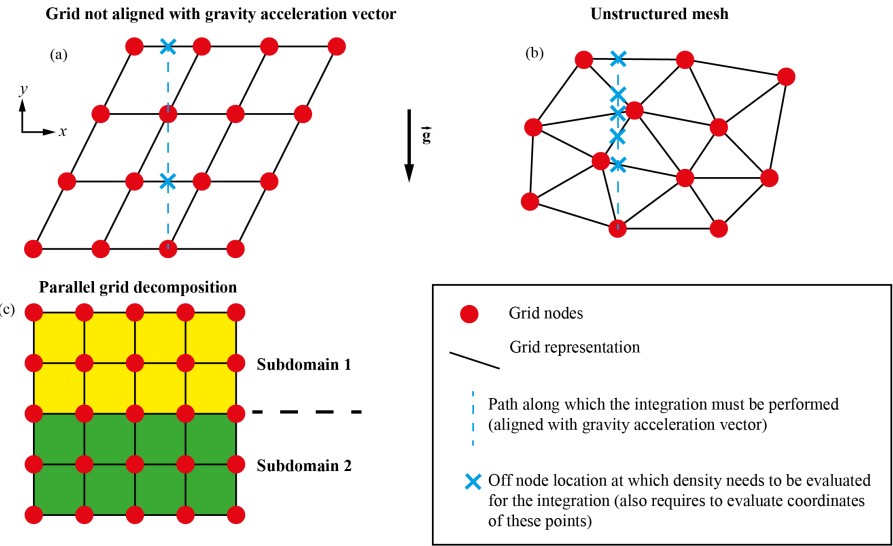

**Figure 1.** Schematic representation of meshes for which computing an integral in the vertical direction can be challenging. **(a)** Mesh with a grid not aligned with the the gravity vector. **(b)** Unstructured mesh. **(c)** Parallel distribution of a mesh. The dashed blue lines represent the direction along which the integral must be performed. The blue crosses represent the points that have to be evaluated during the integration.

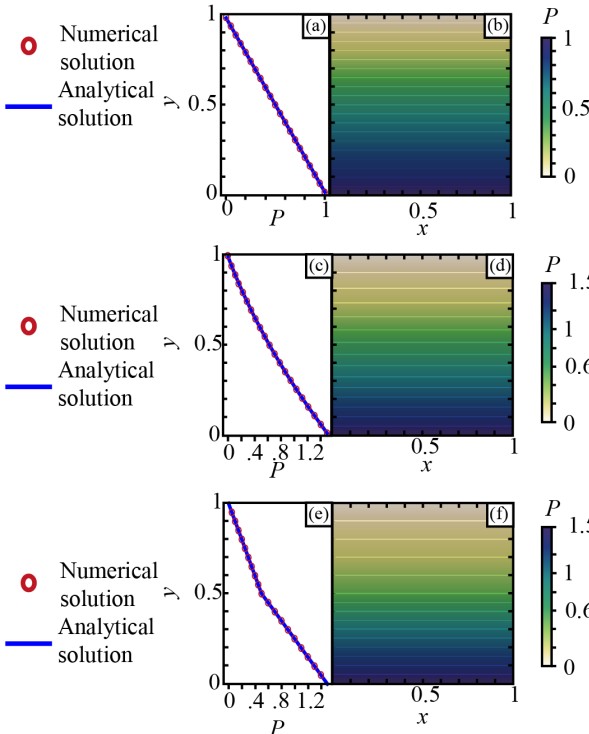

**Figure 2.** Pressure for non-dimensioned hydrostatic cases. **(a)**, **(c)**, **(e)** 1D pressure for **(a)** a constant density $\rho = 1$, **(c)** a continuous $y$ dependent density $\rho = 2 - y$ and **(e)** a discontinuous density. The blue line is the analytic solution computed with Eq. (2), the red circles represent the numerical solution computed with Eq. (7). **(b)**, **(d)**, **(f)** 2D numerical solution for **(b)** a constant density $\rho = 1$, **(d)** a continuous $y$ dependent density $\rho = 2 - y$ and **(f)** a discontinuous density.

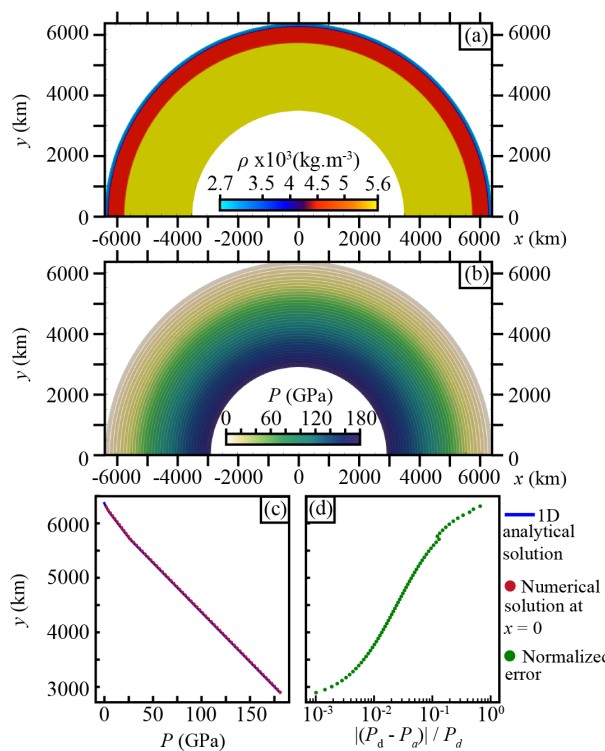

**Figure 3.** Pressure for a hydrostatic case in a half annulus approximating a simplified and idealized layered Earth. **(a)** Density structure for the half annulus model. **(b)** Numerical solution of the 2D half annulus model. **(c)** The blue line shows the 1D analytical solution for the density structure showed in **(a)** along a line parallel to the gravity vector. The red circles show the numerical solution extracted from **(b)** at coordinate $x = 0$ along a line parallel to the gravity vector. **(d)** Green dots show the normalized error between the analytical solution and the numerical solution at $x = 0$ as $\frac{|P_d - P_a|}{P_d}$

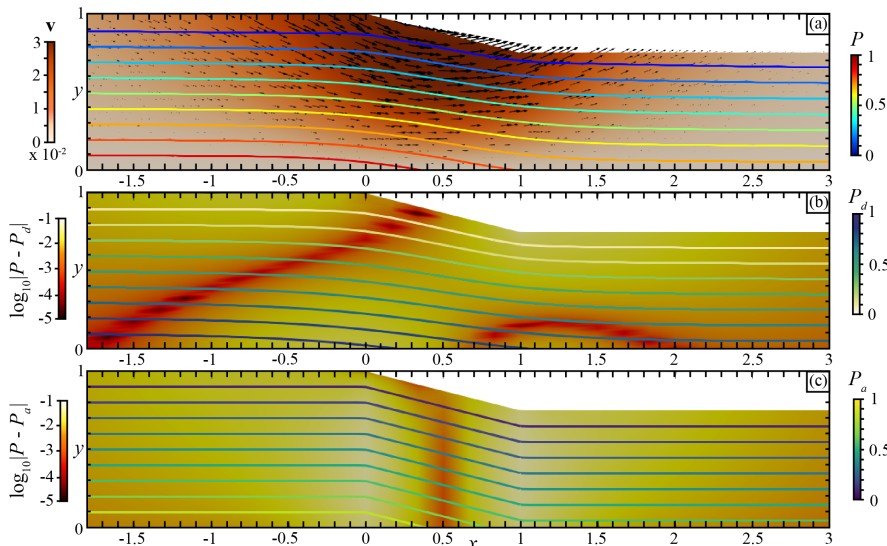

**Figure 4.** Non-dimensional "Global" model for a large domain with a topographic perturbation. **(a)** The background colour shows the velocity field computed with the Eq. (3). The coloured curves show the total pressure $P$ iso-values every 0.1 computed with Eq. (3). **(b)** Comparison between the total pressure from Eq. (3) and the pressure $P_d$ computed from Eq. (7). The coloured background shows the difference $\log_{10}|P - P_d|$. The coloured curves show the pressure $P_d$ iso-values every 0.1 computed with Eq. (7). **(c)** Comparison between the total pressure from Eq. (3) and the pressure $P_a$ computed from Eq. (2). The coloured background shows the difference $\log_{10}|P - P_a|$. The coloured curves show the pressure $P_a$ iso-values every 0.1 computed with Eq. (2).

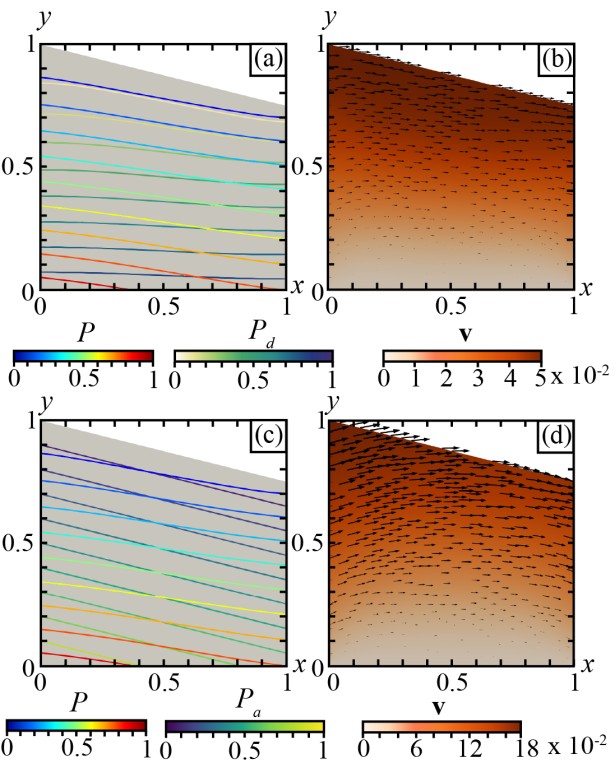

**Figure 5.** Non-dimensional "regional" model for a small domain with a varying topography. **(a)** The coloured curves show the total pressure $P$ iso-values computed with Eq. (3) in the global model every 0.1 and the pressure $P_d$ computed with Eq. (7) in the regional model every 0.1. **(b)** The coloured background shows the velocity field computed with Eq. (3) with stress boundary conditions described by Eq. (33) using $P_d$ computed with Eq. (7). The black arrows show the velocity vectors. **(c)** The coloured curves show the total pressure $P$ iso-values computed with Eq. (3) in the global model every 0.1 and the pressure $P_a$ computed with Eq. (2) in the regional model every 0.1. **(d)** The coloured background shows the velocity field computed with Eq. (3) with stress boundary conditions described by Eq. (33) using $P_a$ computed with Eq. (2). The black arrows show the velocity vectors.

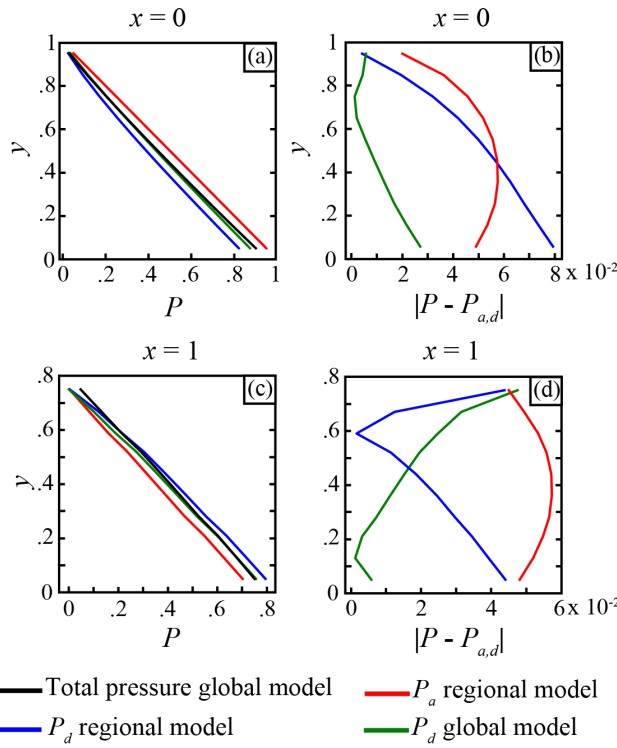

**Figure 6.** Curves showing the pressure from the regional and global models along the profiles **(a)** $x = 0$ and **(c)** $x = 1$. Absolute value of the difference between the total pressure $P$ in the global model and the different pressures in the global and regional models along the profiles **(b)** $x = 0$ and **(d)** $x = 1$.

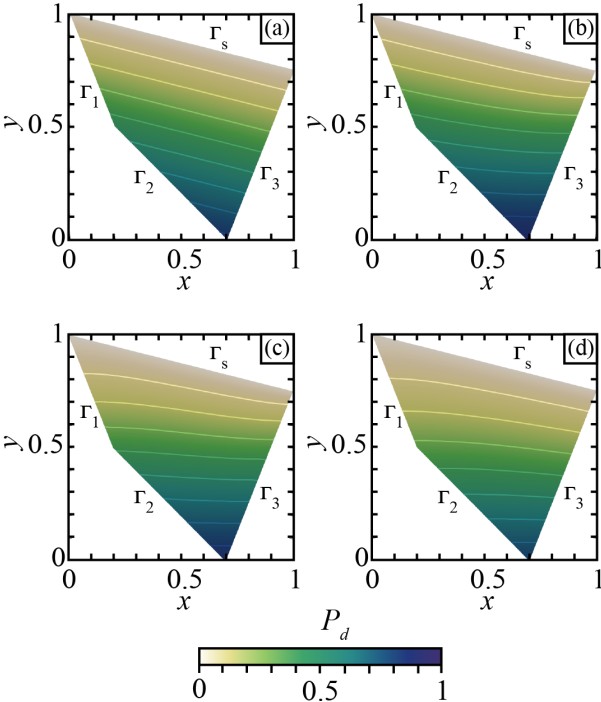

**Figure 7.** Pressure field $P_d$ computed using Eq. (7) for different boundary conditions applied to the individual boundary segments $\Gamma_i, i = 1, 2, 3$. The boundary conditions were defined according to: **(a)** Eq. (15) on $\Gamma_1$, $\Gamma_2$ and $\Gamma_3$; **(b)** Eq. (15) on $\Gamma_1$ and $\Gamma_2$, Eq. (16) on $\Gamma_3$; **(c)** Eq. (15) on $\Gamma_2$ and Eq. (16) on $\Gamma_1$ and $\Gamma_3$; **(d)** Eq. (15) on $\Gamma_3$ and Eq. (16) on $\Gamma_1$ and $\Gamma_2$.

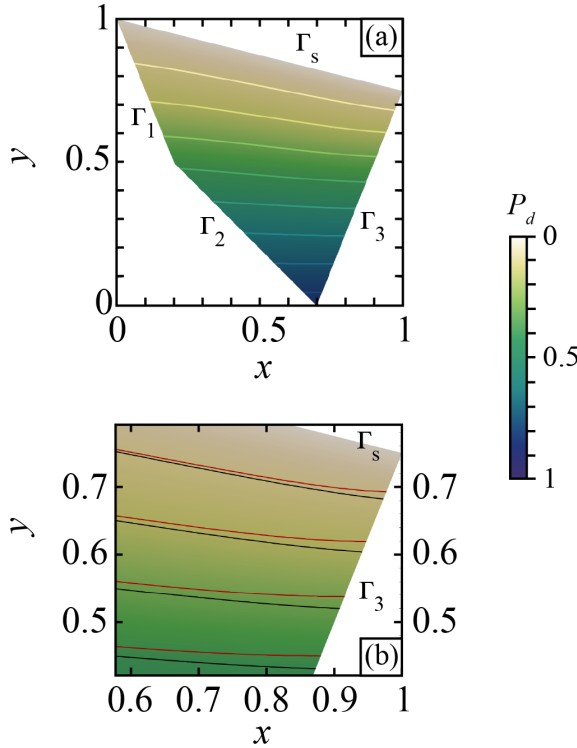

**Figure 8.** Pressure field $P_d$ computed using Eq. (7) with the boundary conditions described by Eq. (23) **(a)** $P_d$ over the entire model with boundary conditions defined by Eq. (23) applied on $\Gamma_1$, $\Gamma_2$ and $\Gamma_3$. **(b)** Zoom on the right boundary region. The black curves show the iso-values of the pressure computed with boundary conditions using Eq. (23). The red curves show the iso-values of the pressure computed using the boundary conditions used in Figure 7c.

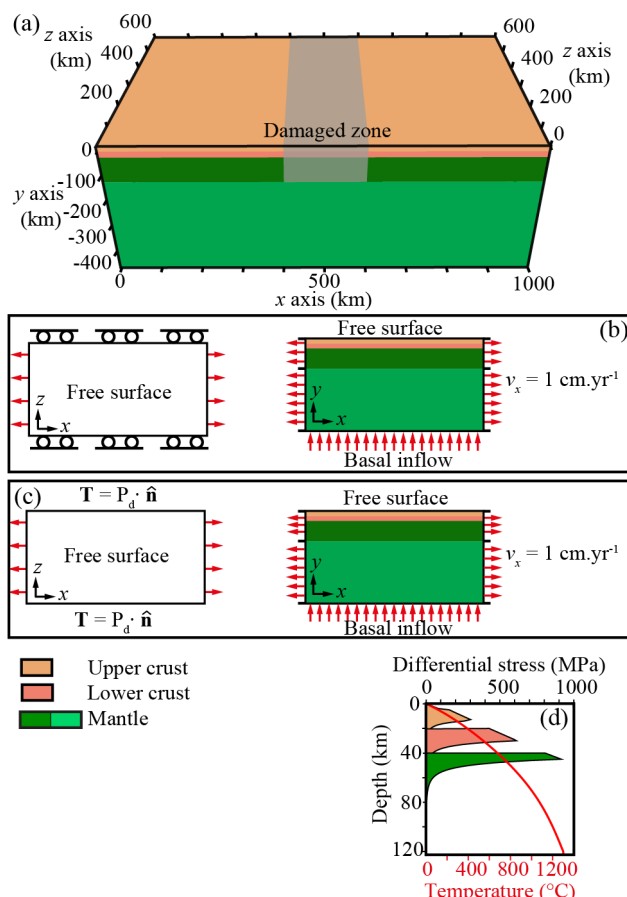

**Figure 9.** (a) 3D view of the modelled domain. An initial plastic strain with a Gaussian repartition is applied in the central part of the domain in the lithosphere. (b) Map and cross-section view of the boundary conditions for the model with free-slip boundary conditions. (c) Map and cross-section view of the boundary conditions for the model with normal stress boundary conditions. (d) Yield-stress envelope and initial temperature of the first 120 km.

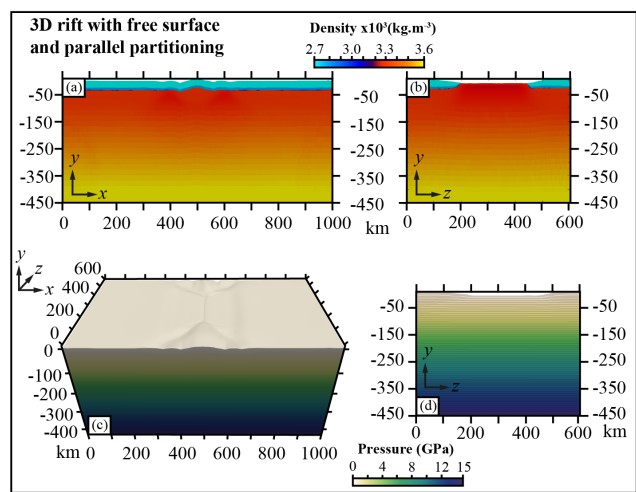

**Figure 10.** (a) Density in cross-section view of the 3D rift model with normal stress boundary condition in the $x - y$ plane at $z = 0$. (b) Density in cross-section view in the $z - y$ plane at $x = 500$ km. (c) 3D view of the pressure computed with Eq. (7). (d) Cross-section view in the $z - y$ plane at $x = 500$ km of the reference pressure. The contour lines are plotted every 0.25 GPa.

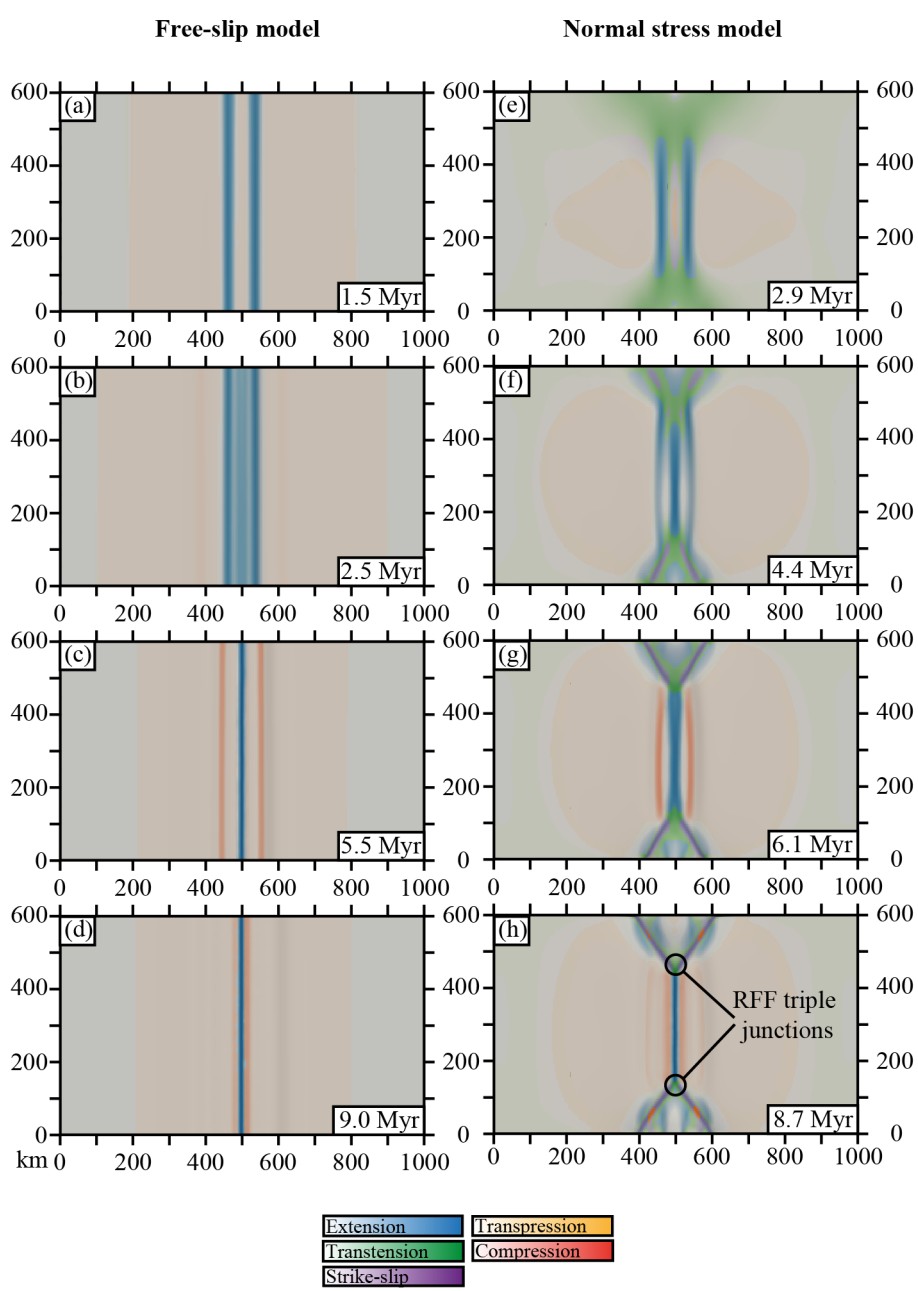

**Figure 11.** Map view of the strain regime evolution in time and space of the model with (a to d) free-slip boundary conditions and (e to h) normal stress boundary conditions.

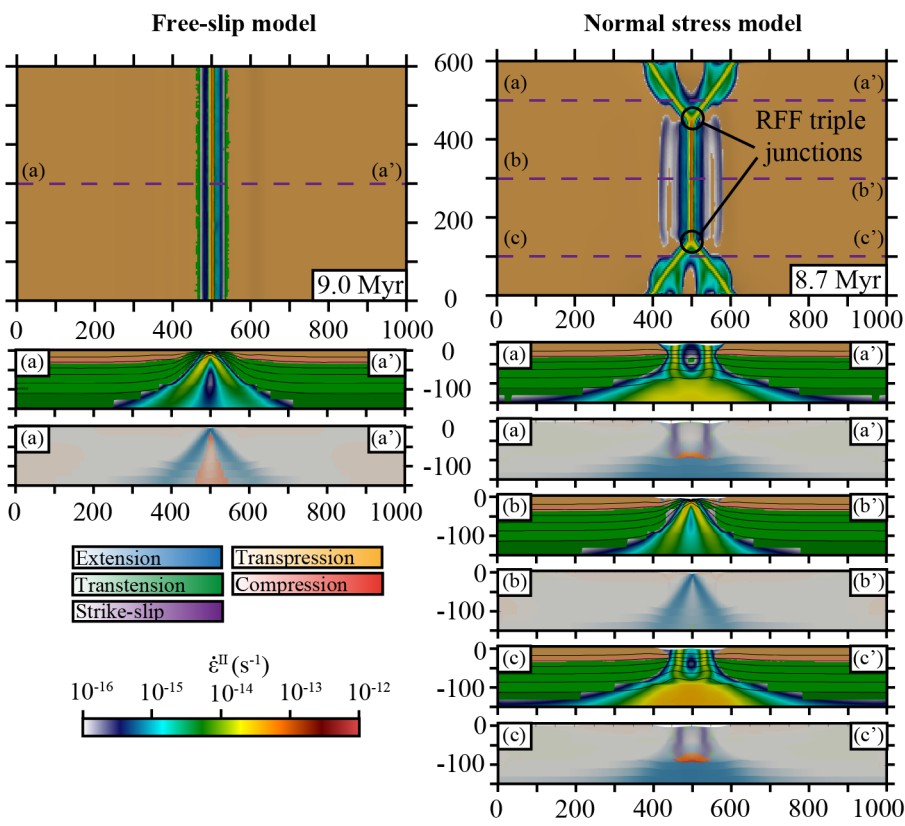

**Figure 12.** Map view of the models with free-slip (left) and normal stress (right) boundary conditions. (a)-(a') , (b)-(b') and (c)-(c') lines indicate the cross-sections displayed below. The cross-sections shows the numerical lithologies with strain rate second invariant (Eq. (36)) and the strain regime.

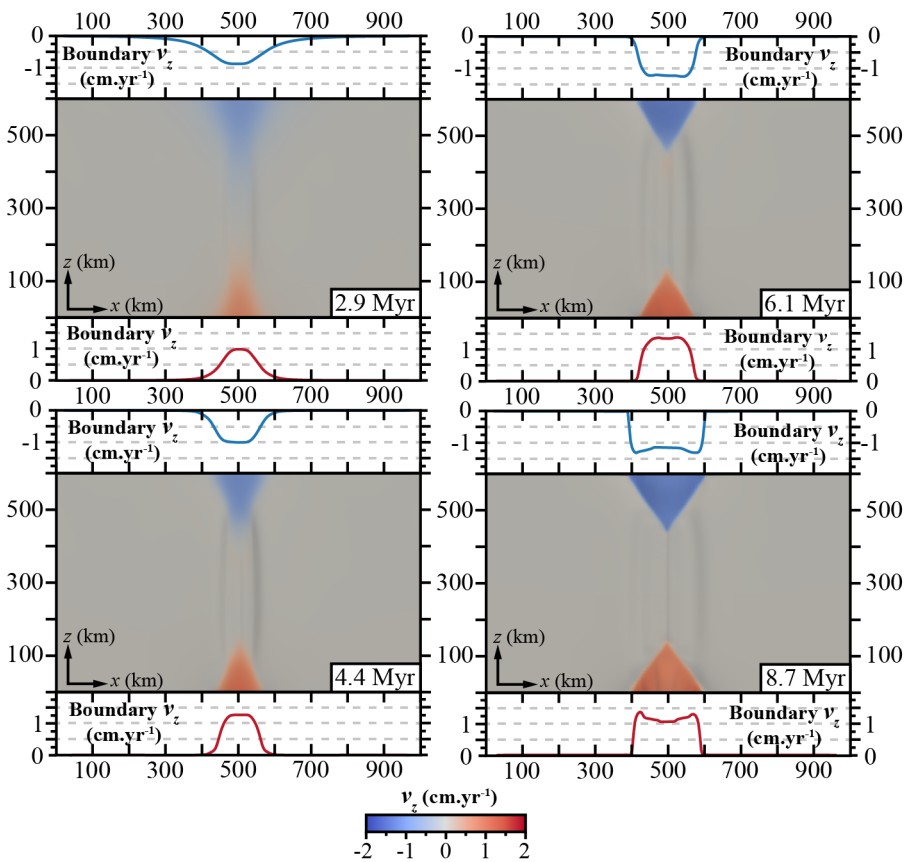

**Figure 13.** Map view of the $z$ component of the velocity ($v_z$). Red curves represent the $z$ component of the velocity along the boundary $z_{max}$ at the surface. Blue curves represent the $z$ component of the velocity along the boundary $z_{min}$ at the surface.

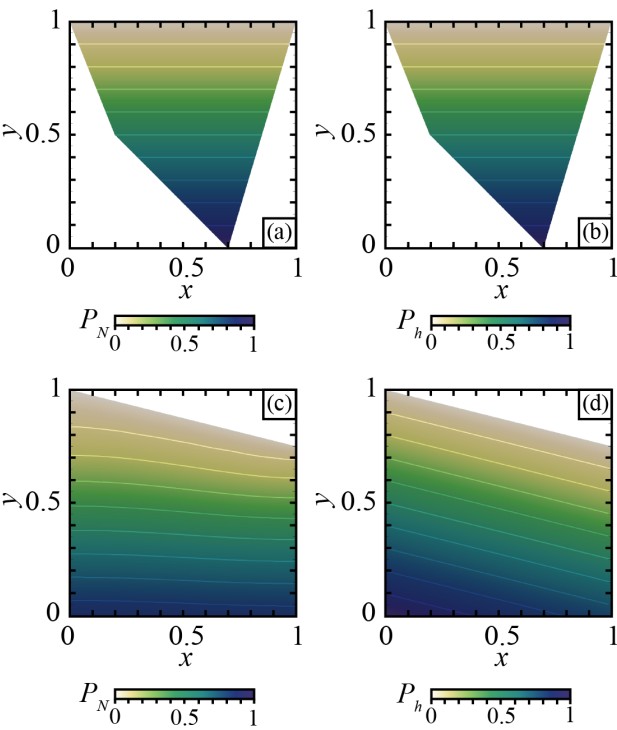

**Figure 14.** Pressure in non-dimensional models with constant density $\rho = 1$. Pressure computed for a hydrostatic case in an irregular shaped domain using **(a)** the normal equations Eq. (45) and **(b)** the hyperbolic equation Eq. (42). Pressure computed for a non-hydrostatic case due to topography using **(c)** the normal equation Eq. (45) and **(d)** the hyperbolic equation Eq. (42). The contour lines show contours of iso-pressure values every 0.1.