# Peer review of "An efficient PDE based method to compute pressure boundary conditions in regional geodynamic models"

_Solid Earth, 2022_

## Author Comment (AC1)

**General modifications**

We are grateful to the reviewers for their questions and remarks. These helped improve the presentation clarity and motivated us to better charaterise the behaviour of the pressure solutions obtained from the proposed Poisson problem. To answer these questions we designed 2D experiments to show the results of the method in hydrostatic and non-hydrostatic cases.

Moreover, we also re-work the boundary conditions for the pressure Poisson problem to include a more general formulation that can be used in arbitrarily shaped domains (it was not the case with the previous formulation).

We also added a discussion section about the different PDE formulations for the pressure depending on the objectives that one might want to achieve.

We added more figures to illustrate all these modifications and discussions in the new version of the manuscript.

Finally, in the tracked change version of the manuscript, we coloured in blue the changes related to R. Gassmöeller remarks and in red the changes related to C. Thieulot remarks. Shown in light blue are changes that were not directly related to reviewers remarks which we thought important to include for clarity and completeness of the study.

**Responses to C. Thieulot**

*The manuscript presents a method which allows to compute the lithostatic pressure in geodynamic models. This method is not new, but it is here clearly explained, as is its advantage over another common approach (the introduction does a great job at highlighting the difficulty/complexity of computing the lithostatic pressure in various common cases). It has also the merit to work in all kinds of geometries. The manuscript is well structured and reads well. The chosen examples speak for themselves.*

☐ *I have quite a few minor comments/questions which I list hereafter. I believe that there is one missed opportunity: the authors do not discuss the case of compressible materials (with potentially with self-consistent gravity) at all. Since it is expected to render the pressure calculation much more complex, and since compressible models are common (esp. in mantle dynamics) I believe this warrants at least a discussion in the manuscript.*

We added a discussion part about the non linearity that arises from compressibility. Lines 489-496.

☐ *line 13: may be add Glerum et al 2018? (https://doi.org/10.5194/se-9-267-2018)*

Glerum et al. 2018 has been added to the list.

☐ *line 20: why not denoting the pressure at the surface $x'_s$ simply $P_s$ and avoid overbars altogether?*

Thanks for the suggestion. We adopted this.

☐ *line 21: overbar missing on $P_0$*

This now appears as $P_s$.

☐ *line 72: it is obvious why the lhs terms of eq3 become zero when $v = 0$, but maybe a short sentence could be added to explain why the deviatoric stress is then also zero.*

We added $\tau = 2\eta\varepsilon(v)$ in the text so it appears that $\tau$ depends on $v$ and hopefully makes it clear why the deviatoric is zero if $v = 0$. Line 79

☐ *line 78: taking the divergence of the momentum equation is indeed common practice in CFD, but this does not justify why the approach is taken here. After all, $\nabla(P) = \rho g$ is a differential equation that could be tackled 'as is'. I think why the*

*divergence approach is necessary should be made clear.*

Taking $\nabla(P) = \rho \boldsymbol{g}$ 'as is' means that there are 3 differential equations but 1 unknown ($P$). If we consider that $\boldsymbol{g}$ is aligned with the coordinate system and $\rho$ is only varying in that same direction then $\nabla(P) = \rho \boldsymbol{g}$ can be reduced to 1 differential equation and 1 unknown but for any other case we need to use the divergence to obtain 1 differential equation and 1 unknown. We added some details about that point lines 91-95. In addition, we also added a discussion in Sec 4.1 to show how a unique solution can be obtained from discretising $\nabla(P) = \rho \boldsymbol{g}$ directly and then by solving the normal equations.

☐ *lines 80+: maybe a short discussion is warranted about the nodes at the corner of the domain? since the intersection of $\partial \Omega_i$ and $\partial \Omega_{surf}$ is zero, these nodes belong to one or the other.*

In domains containing corners, the corners are associated with surfaces where Dirichlet constraints are prescribed. We clarified this point in lines 210.

☐ *line 90: existing->existed*

That part has been removed.

☐ *line 103: why introduce $\boldsymbol{F} = \nabla(P)$ in Eq.11 and never use it further? $\nabla(P)$ could replace $\boldsymbol{F}$ in Eq.10 and it would make the presence of Eq.8 terms more obvious.*

We detailed a lot more the boundary conditions and how to handle the flux term appearing in the LHS in sections 2.1 and 2.2.

☐ *line 114: straightforward*

Corrected line 181.

☐ *line 132: is 'rad' commonly used?*

We removed 'rad'.

☐ *line 125: '2D spherical coordinates' -> polar coordinates?*

Corrected line 223.

☐ *line 126: one usually speaks of the CMB, so core-mantle boundary.*

Corrected line 224.

☐ *line 127: this is a bit unusual. In polar coordinates one would take $\theta \in [0, \pi]$ and $x = r \cos \theta$.*

The notation corresponds to the Figure 2a with the zero centered in the $x$ direction and $y$ positive towards the surface.

☐ *line 131: 'aims showing'*

☐ *line 142: there is a minus sign issue wrt to Eq.3*

We removed the equation from that place as it is now in section 2 and correctly written.

95 □ *line 145: Eq.17 is not needed, simply refer to Eq.4*

Corrected line 340.

□ *line 146: dependent*

100

Corrected line 341.

□ *line 148: in the previous section rho depends on position. If so, it cannot be inserted in the diffusion term to make the heat diffusivity coefficient κ*

105

We corrected that, line 342.

□ *lines 149-150: are the weak forms of the Stokes equations really needed here? they are presented in May et al 2015 about the pTatin3D code. Case in point these equations are not numbered so they are not referred to in the text.*

110

They are written here to make clear from where the Neumann boundary condition comes from.

□ *line 151: is Q1 really used for temperature ?*

115 The way temperature is solved is not really relevant to the manuscript so we just removed this.

□ *line 156: in Eq.19, the exponent should read (1-n)/n or (1/n)-1*

Corrected line 350.

120

□ *line 158: the $\dot{\varepsilon}$ term in Eq.19 is not the second invariant of the strain rate, but rather the square root of the second moment invariant.*

Corrected line 352.

125

□ *line 159: second $\varepsilon_{ij}$ missing in equation*

Corrected line 353.

130 □ *line 161: Although not mandatory, there usually is a factor 2 in the denominator of eq21 (e.g. see eq 7 of Glerum et al) because of the relationship $\tau = 2\eta\dot{\varepsilon}$.*

Corrected line 356.

135 □ *line 172: is it really a Boussinesq approximation if density depends on pressure?*

Yes, because the Boussinesq approximation is for small density variations regardless of the process involved in these variations. We added "The Boussinesq approximation states that perturbations of density, if sufficiently small, can only be considered in the buoyancy term and neglected elsewhere regardless of the origin of the perturbation.", lines 369-371.

140

□ *line 173: $\rho_0$ is not the 'initial' density. It is the density at $T = T_0$ ($T_0$ is missing in Eq.22)*

Corrected line 367.

145 ☐ *line 189: Eq.24 could be written in a more compact form, eg. $\boldsymbol{v} = (1,0,0)$ cm.yr$^{-1}$*

This notation has been removed.

☐ *line 192: mismatch of parenthesis/square bracket*

150

The parenthesis is accepted as a symbolic way to indicate an open interval.

☐ *line 194: the pressure dependence of the density in Eq.22 makes Eq.5 nonlinear but this is not discussed.*

155 Indeed, we account for this non-linearity. It was not stated in text previously, but the reference pressure evaluation and its use as a traction for the Neumann boundary condition occurs at every non-linear iteration for each time step. We added this lines 390-392.

☐ *Aside from this, why is $P_l$ computed only once per time step (and not even at every non-linear iteration?)*

160

$P_l$ (now renamed as $P_d$ for consistency with the new section) is computed at each non linear iteration. This information was missing in the text and has been added lines 390-392.

☐ *Also, prescribing $P_l$ below te Dirichlet b.c. on the $x$ faces echoes the work of Chertova et al 2014 (for example), but I am*
165 *a bit puzzled by what it means to prescribe $P_l$ on the $z$ faces (In the free-slip model, it is akin to say that the model is infinite in the z direction but quid when $P_l$ is prescribed?)*

We added a whole new section with simple models to address the effect of using this pressure as a normal stress boundary condition (section 3.2.2). The free-slip is indeed often considered as "the model is infinite in the z direction", but it also means
170 that the material along the faces exerts an infinite resistance to fluid motion in that direction because flow is prescribed to be zero in this direction and a null resistance to shear (because free-slip also requires a zero shear stress condition). The free-slip condition forces any deformation to be orthogonal to the direction in which the velocity is prescribed to be zero. It can be considered as "infinite" only if the 3rd dimension is an extruded plan and that every displacement and deformation are cylindrical but it is a barrier for non cylindrical deformation and 3D displacements.

175
☐ *line 246: it is*

Corrected, line 500.

---

## Author Comment (AC2)

**General modifications**

We are grateful to the reviewers for their questions and remarks. These helped improve the presentation clarity and motivated us to better charaterise the behaviour of the pressure solutions obtained from the proposed Poisson problem. To answer these questions we designed 2D experiments to show the results of the method in hydrostatic and non-hydrostatic cases.

5      Moreover, we also re-work the boundary conditions for the pressure Poisson problem to include a more general formulation that can be used in arbitrarily shaped domains (it was not the case with the previous formulation).

We also added a discussion section about the different PDE formulations for the pressure depending on the objectives that one might want to achieve.

We added more figures to illustrate all these modifications and discussions in the new version of the manuscript.

10      Finally, in the tracked change version of the manuscript, we coloured in blue the changes related to R. Gassmöeller remarks and in red the changes related to C. Thieulot remarks. Shown in light blue are changes that were not directly related to reviewers remarks which we thought important to include for clarity and completeness of the study.

**Responses to R. Gassmöeller**

*This manuscript discusses a finite-element approach to compute a "lithostatic" reference pressure that can be utilized for*
15 *thermo-mechanical geodynamic models.*

*The computation of a reasonable reference pressure is of large (but often overlooked) importance for geodynamic modeling. Not only can an accurate reference pressure be used to apply open boundary conditions (like proposed in this manuscript), it can also be used to create better reference properties for the equation of state and a better definition of the "dynamic pressure"*
20 *relevant for rheological behavior and comparisons to field studies. I therefore think the topic of the manuscript is relevant and appropriate for Solid Earth.*

*The manuscript has a number of strong points, such as the great care that the authors have taken to make their software and models available and the results of this study reproducible. In addition the authors have found a good balance between*
25 *the mathematical and numerical foundation of the method, numerical benchmarks, and example applications. The figures are visually clear and well labeled.*

**Major points:**

☐   *1. In the current state, there is no evidence in the manuscript that the computed pressure is quantitatively correct. I.e.*
30 *the included benchmarks only provide qualitative figures that show the computed pressure field is similar to what we would intuitively expect. A quantitative comparison to a simple 1D hydrostatic profile, which is mentioned as the main competitive algorithm for computing static pressure profiles, would be easy especially for the sheared rectangle benchmark. But also for a radially layered density profile like the half annulus benchmark a quantitative comparison would be crucial to assess the influence of a discontinuous density on accuracy of the algorithm (see my minor comment below). Finally there is no benchmark*
35 *to illustrate the accuracy of the algorithm for a laterally varying density distribution, like the one in the application model. I would imagine a benchmark like $\rho = \rho(\boldsymbol{x})$, $g_x = 0$, $g_y = const$ in a undeformed box would suffice (I chose this on purpose to illustrate that in this case equation (6) would simplify to laplace $p = 0$, which is exactly the same as for $\rho = const$; is this what you or your readers would intuitively expect?). To have such a benchmark would already shed some interesting light on my second major concern below.*

40

We added two new sections to show the efficiency of the method in hydrostatic and non-hydrostatic cases (section 3.1 and section 3.2). Moreover, we show and discuss the accuracy of the method with respect to the 1D depth integrated method.

□ *2. This brings me to the second large concern, which may be more fundamental than the first. The results of the presented*
45 *method are occasionally unexpected, and there is an inconsistency between what the equation of interest (equation 5) implies*
*(that $\nabla p$ always points in the direction of $\rho \mathbf{g}$) and the solution of the actually solved equation (6) in Fig 2. f) and g). Although*
*gravity is vertical there is clearly a lateral pressure gradient present, which can be seen by the varying depth of the isopressure*
*contours. This inconsistency can also be illustrated by stating the Dirichlet boundary condition that $p = 0$ at the whole outer*
*surface, but allowing that boundary to be non-horizontal. This shows that solving equation (6) is not necessarily the same as*
50 *solving equation (5) and the authors need to make that clear, and explain the constraints of this step. Currently they justify the*
*transformation by stating "Taking the divergence of the momentum equation is common practice when studying iso-viscous flu-*
*ids, and or in the derivation of projection based Navier-Stokes flow solvers (Chorin, 1967)". This is clearly not detailed enough.*
*For example this statement does not explain under which condition this operation is allowed (certain boundary conditions?*
*gravity as the gradient of a potential field? $\rho \mathbf{g}$ equal to the actual gravity force computed from the actual gravity potential?).*
55 *In addition I have read Chorin 1967, it is a 4 page paper that describes a finite-difference algorithm for the Navier-Stokes*
*equation, but I found no reference to taking the divergence of the momentum equation. Is it possible that the authors cited the*
*wrong paper?*

We addressed that remark through several 2D models that are in a non-hydrostatic state specifically due to the presence of a
60 topography (section 3.2). We show the pressure field computed with the 1D depth integrated approach as long as the pressure
field computed with the Laplacian approach (Figures 4, 5, 6). We then compare these results with respect to the total pressure
computed with the Stokes equation to discuss the accuracy of the total pressure approximation of these two methods. We also
show the effect of applying these different pressures as normal stress boundary condition on the flow field. All these results
can be found in the section 3 and on figures 4, 5, 6.
65 In Sec. 2 (around Eq. (6)) we had additional information to motivate why we take the divergence of the momentum equa-
tions. We also provide a different interpretation of the pressure Poisson problem - see Eqs. 8-11. Accordingly we have removed
the reference to Chorin 1967.

□ *3. In my opinion the presentation of the example rift model as a comparison between the authors new lithostatic pressure*
70 *boundaries and free-slip boundaries at the front/back is misleading. The authors have modified nearly every other boundary*
*condition of the model as well (in particular the bottom BC). Changing the bottom prescribed inflow BC into a no-slip boundary*
*condition may well be the reason that more material has to enter the domain through the front/back boundaries (in addition to*
*the now open side boundaries), which could easily lead to the observed change in convective pattern. Therefore it is impossible*
*to tell which of the differences between the two models are due to the use of open boundaries for the front/back, and which are*
75 *simply due to the change in bottom/side boundary conditions. A better comparison would have left the bottom boundary and*
*left/right boundary condition untouched, and only switched the front/back boundary from a free-slip to the lithostatic boundary*
*condition. I agree that for a more realistic model one would also want to change the bottom BC, but not to a no-slip, but*
*to a open/lithostatic boundary (I would consider that an acceptable comparison as well). It may well be the case that with*
*an open/lithostatic bottom boundary (or a prescribed inflow as the reference case) the observed flow would be significantly*
80 *more similar to the reference model. I am afraid this question can only be answered by rerunning the model with the modified*
*boundary conditions, but would urge the authors to do so, since their main conclusion (section 4.1) depends on it.*

We made a new model with normal stress boundary conditions to replace the previous one. This new model uses the same
boundary conditions than the free-slip model except for the faces of normal $z$ on which we apply the normal stress instead of
85 the free-slip condition. The results are indeed different in the asthenosphere because there are no more convection cells (thus
we removed the figure showing the asthenosphere flow since it is no more interesting to show). However, the deformation in
the lithosphere is extremely similar. The strike-slip faults are again developing as long as the triple junctions.

□ *Additional information I would have liked to see:*
90

☐ *I am concerned that the manuscript contains very little information about the limitations of this approach to compute the pressure. In particular the following questions are not answered:*

☐ *- What is the expected accuracy (convergence order) of the algorithm? This is in particular important, because you intend to use the computed pressure field as a boundary condition (as in the application example).*

For the Poisson problem, the convergence order is known to be $\mathcal{O}(h^{k+1})$ in $L_2$ where $h$ is the mesh size and $k$ the polynomial order of the FE approximation. Moreover, in `pTatin3D` we use $Q_1$ elements overlapping the $Q_2$ mesh. So the pressure from the flow problem and the approximated pressure from the poisson share the same convergence order which is $\mathcal{O}(h^2)$ in $L_2$. The RHS of the pressure Poisson problem also requires a projection from particles to quadrature points. This projection is done using a bilinear approximation, which is the same projection used for the density to compute the flow problem. This projection does not affect the order of convergence.

☐ *- You mention that the weak formulation is valid for a discontinuous density, but is it expected to affect the accuracy of the solution?*

We addressed this point in section 3.1 for hydrostatic cases. We specifically made a discontinuous density model. line 213-218: "Since the $P_2$ FE approximation contains the monomials 1, $y$ and $y^2$, the FE solution exactly reproduces the analytic solution for case 1 and case 2 – independent of the number of finite elements used in the domain (e.g. sub-dividing the box into two triangles would be sufficient to obtain an exact solution). For case 3, the analytic pressure solution is piecewise linear, hence provided the density discontinuity is exactly resolved by the faces of the triangular FE mesh (which was the case here), the FE method exactly reproduces the analytic solution."

☐ *- Is it important that $\boldsymbol{g}$ is the gradient of a potential field? In reality that will always be the case, but in numerical models, in particular benchmarks it may not be (e.g. it may be a purely rotational vector field).*

No. No part of our formulation requires or assumes that $\boldsymbol{g}$ is the gradient of a potential.

**Minor comments:**

☐ *- lines 48-49 the sentence is missing a verb, or 'if' should be 'of'*

Corrected line 56

☐ *- line 90: 'if there' seems wrong*

That part has been removed

☐ *- line 105: The current reference to equation (10) is ambiguous (does it reference the BC or the surface integrals?). The definition of the boundary condition happens in eq (8). Either reference eq (8), or reword to: "Furthermore, from the definition of the boundary conditions the two surface integrals on the LHS and RHS of equation(10) cancel.*

This part has been modified to better develop the general boundary conditions for any geometric case (*i.e.* for domains with arbitrary boundaries), section 2.1

☐ *- line 125: this is usually called the 'polar angle' or 'azimuth'. 'angle' is ambiguous.*

Corrected line 223

☐  - *line 125: I understand that you only provide an example model, but quoting the Earth's radius as 6375 km and the depth of the core-mantle boundary as 2700 km without qualification is extremely inaccurate. The canonical value for an averaged spherical Earth radius is 6371 km (no matter the exact definition), and the depth of the core-mantle boundary is 2891 km (+/- a few km depending on source). Either qualify that you use simple values for illustration purposes or correct the values.*

We changed for the values proposed. But we also state that we provide an approximation, lines 222-225

☐  - *line 132: This sentence is grammatically not correct: "aims showing" -> "intends to show"*

We removed that example in favour of a more detailed study for a deformed domain in a non-hydrostatic case.

☐  - *equation (18) is written in a slightly unusual form in that the equation was divided by $\rho C_p$ and the factor was incorporated into the the thermal conduction term to form the thermal diffusivity. This is strictly only possible if the density and specific heat capacity are spatially constant. In many simplifications of the temperature equation this is actually the case (e.g. the Boussinesq or the Anelastic Liquid Approximation), but it is unclear if you used these. Additionally the heat source H in the equation seems to be the volumetric heat source, while typically the term is written as $\rho H$ with H being the specific heat source (which is easier to determine for rocks). Please clarify these terms, or use a more conventional form of the equation (e.g. eq. 6.10.49 on pp. 273 of Schuberth,Turcotte,Olson "Mantle Convection in the Earth and Planets").*

We changed the form of the equation to indeed match the equation we actually solve line 342.

☐  - *eq(20) does not include the definition of the second invariant as it claims to do, it only specifies the square root and factor $\frac{1}{2}$, but it does not specify how to convert the tensor into the second invariant*

The initial formulation missed one term. It has been corrected. As well as we added in the text that we take the square root of the second invariant. Line 352-353

☐  - *line 167-168: "takes place" and "lays" are weird formulations to describe something that exists/extends. I suppose you tried to avoid repetitive use, but if there is one word that describes what you want to say, use it repeatedly instead of replacing it with less precise versions. The same holds true for "modelled" vs "simulated" in the same paragraph. Using the same words will improve the readability of this paragraph.*

Corrected line 361-362

☐  - *eq(22) is actually an extension of the original Boussinesq approximation. The original BA explicitly neglects density changes due to pressure. Since these changes are typically at least an order of magnitude smaller than density changes due to temperature this will not affect your models much, but you can not claim to precisely implement the Boussinesq approximation here. Seeing this equation also raises the question which density you used for the temperature equation? The BA requires to use $\rho_0$ the reference density, but in Fig. 2. d) you show a density that increases with depth.*

We corrected the equation. We also corrected the temperature equation to show that we indeed use $\rho_0$ (line 342). As for the Stokes equation, we use the Boussinesq approximation, *i.e.* the density variations due to pressure and temperature are only accounted in the buoyancy forces term. Nevertheless, we also use the density variations to compute the pressure approximation in the pressure Poisson equation. We also added more details about the Boussinesq approximation to the text in the manuscript: "The Boussinesq approximation states that perturbations of density, if sufficiently small, can only be considered in the buoyancy term and neglected elsewhere regardless of the origin of the perturbation." (lines 369-371)

□ - *eq(23) This is an unusual choice as initial condition for a lithosphere model. A steady-state solution will be a conductive profile across the whole domain (down to a depth of 450 km), while in real models everything below the lithosphere will be convecting sufficiently to create an adiabatic temperature profile following the average mantle potential temperature. This convection would lead to much higher temperature at the boundary between lithosphere and asthenosphere and could*
190 *therefore significantly change the strength of the lithosphere in your model. For a science application this would be crucial to correct, but since you here only show the difference between the boundary conditions it is likely ok. However, you should at least mention that this is a simplification of a realistic profile.*

Indeed, we forgot to mention that we do simulate an adiabatic gradient for the initial temperature. We added it in the text
195 line 374-377: "Moreover, to simulate an adiabatic thermal gradient in the asthenosphere due to thermal convection, the initial temperature field is solved with a conductivity of $k = 70$ W.m$^{-1}$.K$^{-1}$ in the asthenospheric mantle. However, to avoid preventing convection in the asthenosphere during the time-dependant simulation, a conductivity of $k = 3.3$ W.m$^{-1}$.K$^{-1}$ is used to solve Eq. (34)."

200 □ - *Section 4.1: Since you only have a single subsection in the discussion, do not introduce 4.1., instead reword the heading of "Discussion"*

We added more discussion parts.

205 □ - *Discussion and Conclusions are very brief. In particular these should contain a reference if these new patterns of deformations are also observed on Earth, what kinds of applications are additionally available through your new method, and what kinds of limitations or challenges remain.*

The discussion in now more detailed. There are references to studies that produced 3D geodynamic models with results that
210 are relatively similar. However, in this paper the 3D model with normal stress boundary conditions is mainly a demonstrator. We did not intend to reproduce an actual geodynamic system on Earth.

□ - *line 247 "amenable parallel computing environmnents" is missing a "to", however what you really want to say is probably "applicable in"*
215
Corrected line 500

□ - *line 253 - 255 this sentence is too long and complicated and you forgot at least one "that". Split the sentence to make the argument easier to follow.*
220
Corrected line 510

---

## Author Response (AR2)

**1   C. Thieulot**

*The manuscript has undergone a major overhaul is has been substantially improved. I recommend publications after some minor corrections be made:*

*eq 8,9: tau should be bold*

Corrected in Eqs. (8) and (9).

*l117 eq 12, not 15 ?*

Indeed, we corrected that line: 116

*l 120: I may have missed it but I don't see a definition of $\hat{t}$*

We added: "with $\hat{t}$ a tangent unit vector to the boundary such that $\hat{n} \cdot \hat{t} = 0$." line: 120

*l 121: eq 15 or eq 12?*

Eq. 12, we corrected that line: 121

*l 148: do not provide*

Corrected line 147

*l 178: eq 28 or eq 25?*

We keep Eq. (25) as it is the general form before applying any choice for the Neumann boundary conditions.

*eq 34: missing parenthesis*

Corrected line: 336

*eqs in 3.4.1: dot on $\varepsilon$, but not in section 2*

We added the dot in section 2 for coherence.

*l 470: wither*

Corrected line: 461

*fig 4: colorscales for Pa and Pd are inverted*

We modified the figure so all the models share the same colour scale.

**2 R. Gassmöeller**

*Thank you for thoughtfully addressing my questions and concerns. I think the manuscript is now in a much better state and gives reader a much clearer picture about the benefits of using the poisson pressure equation. Due to the large amount of additional material in the manuscript I have accumulated a somewhat lengthy list of minor comments that I would like to see addressed. Even my one major comment is mostly a request to clearly state the conclusion of your new sections 2.1./3.3. If you carefully address my remaining comments I am happy to suggest this manuscript for publication.*

**2.1 Major comment:**

*- I appreciate that the boundary conditions for equation (7) are now spelled out in more detail in section 2.1 (lines 119-154), however I think the current description leaves the reader more confused than enlightened. You split the Neumann boundaries in two categories (omega perpendicular and omega parallel). But you do not discuss why you do that, or what happens if your side boundaries were not parallel/perpendicular to the gravity field. You also do not explain how you distinguish boundaries into the two categories. Next, your split version of the Neumann BC has some nonintuitive properties that you discuss in detail, to conclude at the end of the paragraph that equation (23) is a single version of the BC that would give reasonable results. Indeed equation (23) seems perfectly reasonable to me, and I would have expected that to be the Neumann BC for all boundaries. So at the moment I am at a loss why you had to introduce the complications before (and the reader willprobably be too). There may be a simple explanation, I just think it is important to spell this out in the manuscript.*

*Later comment: After reading section 3.3. I understand this issue better, in particular that there are multiple possible choices for the Neumann BC, and that results between them differ (I think this should already be pointed out in Section 2.1, you can refer the reader to the later section for the results). But I still do not understand why you chose equation (15) and (16) instead of choosing (23) for everything. I think section 2.1 requires a sentence similar to the following: "We chose to use eq.(15) and (16) as Neumann boundary conditions for the following reasons .... ." If you chose (15),(16) for technical reasons, add something like: "If possible we suggest using equation ... as the best option for Neumann BC for the PPE problem." or some other qualification. In its current form this section is missing a concluding point that the reader can take away from the discussion of the different options.*

We added "Nevertheless, for an arbitrarily shaped domain, using boundary conditions Eq. (15), (16) or (23) does not yield the same result (see section 3.3). For a general use (i.e. when considering arbitrarily shaped domains) we suggest to employ Eq. (15) & (16) as they are a direct extension of the 1D hydrostatic assumptions to 2D and 3D domains."
Lines: 153-156

**2.2 Minor points:**

*- line 67,68: "it is still important to approximate the total pressure in(!) the best possible way"*

Corrected line: 68

*- line 79: you now explain tau using eta, which itself is not defined. Add an explanation for that.*

Corrected lines: 79-80

*- line 84: "along which" seems wrong*

Corrected line: 83

*- line 207/208: Here you state that you use the equation for $\Omega_\parallel$ for the base of the domain and $\Omega_\perp$ for the sides of the domain. This suggests parallel and perpendicular are referring to the normal of the boundary, not the boundary itself. This makes sense, but was not explicitly stated before, and I intuitively expected it to be the other way (in 2D). Please add an explanation for this in line 126 so that readers are aware of it from the start. Also line 126 is missing a partial derivative for the parallel boundary.*

We added the missing partial sign here (line: 126). The parallel and perpendicular signs are not defined with respect to the boundary or the normal to the boundary, they are defined with respect to the gravity vector. $\partial\Omega_\perp$ defines the boundaries on which the constraint $\nabla P \cdot \hat{\boldsymbol{g}}_\perp = 0$ is applied while $\partial\Omega_\parallel$ defines the boundaries on which the constraint $\nabla P \cdot \hat{\boldsymbol{g}} = \rho g$ is applied. It is already stated by equations (15) and (16) under the form for all $\boldsymbol{x} \in \partial\Omega_{\perp/\parallel}$.

*- line 220: "aims showing" seems wrong*

Corrected line: 218

*- line 233: "very small": you give a good explanation for why the solution in spherical coordinates is not as accurate, but please quantify "very small" either in absolute (x MPa) or relative terms (are we speaking about 1% different or 1e-6 difference?). This question is relevant if you want others to adapt your method to give them a way to quantify if they implemented your method correctly.*

We added a plot on Figure 4 (d). We also refer to that plot in the text line: 231

*- Fig. 4.: This is a great figure to show the difference between the two approaches on a well chosen example. I have one problem with the current visuals however: The colorscale in plot b) and c) emphasizes regions with very small errors (yellow) and deemphasizes regions with very large error (black). It took me more than a minute to figure out that your new method is indeed better, because I thought clearly plot b) shows larger errors for your new method. Consider flipping the colorscale I think that would greatly help the visual clarity of the figure.*

We flipped the colour scale of Figure 4.

*- line 283: The Neumann BC described here are correct (because you use a cartesian box), but earlier you always describe them relative to the normal of the boundary, while here you describe them using g and $\hat{\boldsymbol{g}}$. Wouldnt it be more consistent to stick to describing them using the boundary normal?*

We stated the boundary conditions with respect to $\hat{\boldsymbol{n}}$ lines 278-279

*- line 293: "On Figure ..." -> "In Figure ..."*

Corrected line: 288

*- line 322: I think you mean "force the pressure gradient to be parallel (!) to g"*

Indeed, corrected line: 316

*- line 325-327: I think this statement (PPE=depth integrated approach for certain BC) is only true for constant gravity and density fields as in this model. Please clarify in manuscript.*

No, it is also true for a varying density field. We will not show this on a figure but it is easily verifiable with the firedrake scripts we provided. Taking the box model and setting $\rho = 1 + x$ you will see that using the BCs from Eq. (15) on faces of

normal x and Eq. (16) on the bottom face will produce the same solution than applying the 1D approximation.

*- line 342: thank you for adapting the equation as I requested, however now you are missing parentheses around the material derivative on the left-hand side of the equation.*

Corrected line: 336

*- line 376-377: This phrasing is somewhat unusual. You do not really use k=3.3 to avoid preventing convection, you use it because it is a realistic value for the thermal conductivity. 70 was previously just used to simulate convection and an adiabatic temperature gradient. Rephrase maybe to: "However, for the actual model run we used a more realistic conductivity of ...".*

Corrected line: 370

*- line 428: "solve" -> "sole"*

Corrected line: 420

*- Fig. 14: Please clarify in the Figure caption which color scale refers to which feature in the figure, they are sufficiently similar to make that hard to see. Also the figure caption does not make clear what the background color shows vs the isocontours, the caption only mentions a single quantity for each figure. Are the contours just contours of the background color? Also why is the color scale flipped between the two columns? To make clear that one is Ph vs PN? This could be made clear with a label. Using the same color scale would make comparisons easier.*

We modified the figure so all models share the same colour scale. We also indicate that the contour lines represent iso-values of pressure every 0.1.

*- line 470: "wither" -> probably "whether", also this sentence is pretty complex with multiple parentheses and seems to be missing a final statement, or I do not understand the structure of this sentence. Please simplify.*

We corrected whiter to whether, removed the parentheses and added (i) and (ii) at the beginning of the possible choices. Line: ??

*- line 473,474: "Therefore" and "thus" are duplicative*

We removed thus, lines: 461-462

*- Line 478: You now discuss a lot of other topics since presenting the geodynamic lithosphere model. Clarify which model you are talking about in this section.*

We added at the beginning of the section: "In the geodynamic rift model". We also added references to figures illustrating what is stated in the text to avoid any ambiguity. Lines: 469-471

*- Section 4.4.: You state that this equation could become nonlinear, but you do not discuss what to do about it. Do you suggest using the reference quantities to avoid nonlinearities since it is just an approximation to the real pressure, or do you suggest to include the nonlinearity and handle it somehow?*

Actually in the geodynamic model section we state that we evaluate the PPE using a non-linear solver and that we apply its value as a boundary condition at each non-linear Stokes solve iterations. We do not specifically suggest anything, it is up to modellers to choose if they want to simplify the non-linear problem to a linear one, or to solve it as it is.

*  - line 506: demonstrates -> demonstrate*

Corrected line: 496